# Efficiently Robust In-Context Reinforcement Learning with Adversarial Generalization and Adaptation

**Juncheng Dong**[*]    **Hao-Lun Hsu**[*]    **Miroslav Pajic**[†]    **Vahid Tarokh**[†]
Duke University
{juncheng.dong, hao-lun.hsu, miroslav.pajic, vahid.tarokh}@duke.edu

## Abstract

Transformer models (TMs) pretrained on diverse datasets exhibit impressive in-context learning (ICL) capabilities, enabling them to adapt to new tasks without parameter updates. In reinforcement learning (RL), in-context RL (ICRL) leverages this capability by pretraining TMs on diverse RL tasks to adapt to unseen ones. However, the robustness of pretrained TMs in ICRL remains underexplored, with performance often degrading under disturbances in deployment. To address this, we propose a pretraining framework that augments the data with adversarial variations of training environments. This approach improves robustness and enhances generalization by increasing task diversity, while avoiding the computational overhead and pessimism of worst-case optimization used in robust RL. We further introduce an adaptive variant that uses the ICRL capability of pretrained TMs to efficiently generate high-quality data through online rollouts. Extensive experiments on a diverse set of ICRL tasks prove the efficacy of the proposed methods.

## 1 Introduction

Transformer models (TMs) pretrained on a massive amount of data have achieved remarkable successes spanning a wide range of application areas [1, 24, 37]. Notably, TMs exhibit remarkable in-context learning (ICL) capabilities, solving *unseen* supervised learning tasks using only a few input-label pairs as demonstrations, *without requiring any parameter updates* [28]. Although reinforcement learning (RL) represents a more complex sequential learning problem than supervised learning, *in-context reinforcement learning* (**ICRL**) has recently found impressive achievements. ICRL methods first pretrain TMs on a diverse set of RL tasks and subsequently deploy these pretrained TMs to new environments (tasks) unseen during the pretraining.

One particularly successful pretraining method is through supervised pretraining which directly trains the TMs to predict the optimal actions for some query states across different RL tasks, conditioned on a given *context dataset* consisting of trajectories collected from those pretraining environments [16]. See Figure 1(a) for the supervised pretraining framework. During **pretraining**, TMs learn to infer the optimal policies by leveraging the environmental information embedded within the trajectories of the given context dataset. During **deployment**, the pretrained TMs act as policies for new RL tasks, with only a couple of trajectories collected from the new RL tasks as the context dataset to condition on. See Section 2 for details of supervised pretraining and Algorithm 1 in Appendix H for deployment.

**Challenges.** Although pretrained TMs exhibit impressive performance by inferring near-optimal actions in new RL tasks, robustness has yet to be thoroughly examined. We observe that even mild perturbations(representing corrupted or unreliable interaction data) can lead to *significant performance degradation*.

---

[*]Equal contribution
[†]Equal advising

39th Conference on Neural Information Processing Systems (NeurIPS 2025) Workshop: Reliable ML from Unreliable Data.

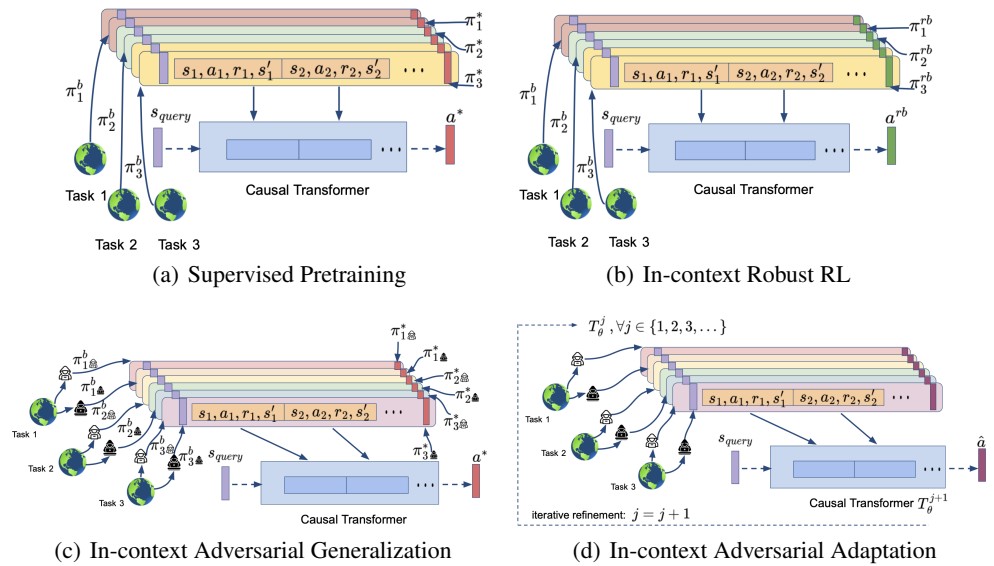

(a) Supervised Pretraining      (b) In-context Robust RL

(c) In-context Adversarial Generalization      (d) In-context Adversarial Adaptation

Figure 1: Overview of the supervised pretraining framework for ICRL across multiple tasks, where each task corresponds to an MDP instance $\tau^i$, with $\tau^i \in \{1, 2, 3\}$ as illustrated. **(a)** A TM is pretrained to predict the optimal action across RL tasks [16], using optimal action labels $a^\star$ from expert policies $\pi^\star_{\tau^i}$, given query states $s_{\text{query}}$ and offline trajectories collected from different RL tasks by distinct behavioral policies $\pi^b_{\tau^i}$. **(b)** To enhance policy robustness, a robust action label $a^{rb}$ is sampled from the robust policy $\pi^{rb}_{\tau^i}$ rather than $\pi^\star_{\tau^i}$. **(c)** To approximate robust learning without explicit max-min optimization, each task $\tau^i$ is augmented with $K$ adversarial variations $\{\phi^{i,k}\}_{k=1}^K$ (e.g., black and white hackers), and the optimal action labels are derived from $\pi^\star_{\tau^i, \phi}$, representing the optimal policy under disturbance $\phi$. **(d)** Alternatively, high-quality action labels $\hat{a}$ and trajectories are generated for $K$ task variants via recursive online adaptation using pretrained models $T_\theta^j$, for $j \in \{1, 2, 3, \dots\}$.

In Figure 2, we present experiments in the Dark Room environment [15] to investigate the performance of pretrained TMs under deployment disturbances. Deployment disturbances can be viewed as corrupted interaction data or adversarial manipulations in the RL environment, leading to unreliable observations and degraded performance. Our results show that their performance degrades under even mild disturbances, specifically with action perturbation applied at probability $p = 0.2$. Although the magnitude of degradation varies across methods (i.e., DPT [16], prompt-DT [43], mix-DPT [39]), the consistent downward trend reveals a common vulnerability: *While ICRL models are pretrained to adapt to new tasks or environmental changes, they still exhibit limited robustness to perturbations*—an important challenge in real-world applications. We attribute this observed lack of robustness to distributional shift, that is, the environmental change caused by perturbation is out-of-distribution with respect to the ICRL pretraining tasks.

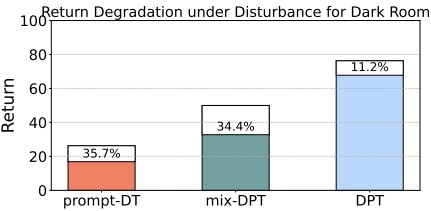

Figure 2: Return degradation of three in-context RL methods under action perturbation with probability $p = 0.2$ in the Dark Room environment. Bars represent performance without disturbance, while the hatched white regions indicate the portion of return lost due to disturbance.

This observation motivates us to design an ICRL framework **(a)** with *robust performance* under disturbance while **(b)** maintaining *strong generalization* to new tasks. To address this challenge, we draw inspiration from solutions developed for standard RL. In these settings, one of the most effective and popular approaches to improve policy robustness is through **worst-case optimization**, that is, to find policies with the best worst-case performance among an uncertainty set through solving a

max-min problem [22, 26]. In the sequel, we will refer to these policies obtained through max-min optimizations as **robust policies**.

Notably, within the supervised pretraining framework for ICRL, the worst-case optimization approach from standard RL can be directly extended to ICRL by simply modifying the action labels used during pretraining. See Figure 1(b) for illustrations and Section 3 for details. In this case, *we pretrain TMs to infer robust actions* that would be taken by robust policies in new RL tasks.

However, this approach has several limitations. **First**, the worst-case optimization approach is known to have two challenges: *(a)* its success hinges on the quality of its uncertainty set construction, e.g., the worst-case formulation can easily to lead to over-pessimistic policies if the uncertainty set includes too many environments [6]; *(b)* in the current era of deep RL, characterized by complex policies and value estimators, the associated max-min problems are often difficult to be solved efficiently. **Moreover**, while solving the worst-case optimization problem for a single RL task is already challenging, identifying robust policies across a large and diverse set of RL tasks is even more demanding.

**Contributions.** Our insight for addressing these challenges is that we can use the in-context learning capability of TMs to transform the robust optimization (i.e., max-min) problems into generalization problems. Instead of pretraining the TMs to infer a robust policy for a new RL task, we propose to *directly* pretrain them to *generalize* across varying disturbances and adversaries. We term this proposed approach **In-context Adversarial Generalization (ICAG)**.

Thus, the proposed framework ICAG requires generalization on two axes. One axis is the generalization across different RL tasks; the other axis is the generalization across varying disturbances and adversaries, thus addressing the out-of-distribution problem. To help TMs achieve generalization on these two axes, we propose to *augment* the pretraining dataset to include perturbed environments. Specifically, for every pretraining task, we create multiple variations of it, each of which including a randomly sampled adversary. We learn an optimal policy for every variation, and use it to generate action labels for supervised pretraining. In particular, the adversaries remain fixed and is deemed as an integral part of the environment so that identifying the optimal policies for variation environments can be parallelized and considerably easier than finding a robust policy through max-min optimization. See Figure 1(c) for a conceptual demonstration.

ICAG offers many notable benefits. From the perspective of improving policy robustness for standard RL, ICAG *avoids sacrificing performance for robustness*, as it can adapt to varying environments and disturbances while robust policies often show degraded performance due to challenges such as over-pessimism and effective solutions to the max-min problems. In particular, the adaptation is in-context and *without any parameter update*. This provides considerable practical advantages as we only need to deploy one fixed model, especially in the common scenarios where active updates of models are expensive or even infeasible. Furthermore, we observe that ICAG also improves ICRL performance when deployed to new environments *without* disturbances. This is because ICAG also functions as a data augmentation method, where the generated disturbances effectively *increase the diversity* of pretraining tasks to help generalization

To further reduce the computational burden of collecting optimal action labels for each variation environment, we introduce **In-context Adversarial Adaptation (ICAA)**, a complementary method to ICAG. Instead of solving for the optimal policy from scratch, ICAA efficiently adapts a pretrained ICRL model to each variation environment through online rollouts. It then extracts high-quality query state–action pairs from high-performing trajectories to augment the pretraining dataset. This recursive process iteratively enhances the robustness and generalization capabilities of the model with minimal training overhead, which is illustrated in Figure 1(d).

Throughout extensive experiments on sparse-reward navigation tasks [15] and two complex continuous control tasks (Meta-World and MuJoCo with 11 problems [36]), our ICAG and ICAA demonstrate superior performance, particularly when under agent disturbances during deployment.

## 2  Preliminary

**Markov Decision Process (MDP).** Sequential decision-making tasks are modeled as MDPs [12, 2]. An MDP $\tau$ is defined by the tuple $(\mathcal{S}, \mathcal{A}, P_\tau, R_\tau, \gamma, \rho_\tau)$, where $\mathcal{S}$ is the set of states, $\mathcal{A}$ is the set of actions, $P_\tau : \mathcal{S} \times \mathcal{A} \to \Delta(\mathcal{S})$ is the state transition function, $R_\tau : \mathcal{S} \times \mathcal{A} \to \mathbb{R}$ is the reward

function, $\gamma \in (0,1)$ is the discount factor, and $\rho_\tau \in \Delta(\mathcal{S})$ is the initial state distribution. At each time step $h$, an agent selects an action $a_h \in \mathcal{A}$, receives a reward $r_h = R_\tau(s_h, a_h)$, and transitions to the next state $s_{h+1}$ according to $P_\tau(s_h, a_h)$. A policy $\pi : \mathcal{S} \to \Delta(\mathcal{A})$ maps the current state to an action distribution. The agent's goal is to learn the optimal policy $\pi_\tau^\star$ that maximizes the expected cumulative reward $G_\tau(\pi) = \mathbb{E}[\sum_{h=1}^{\infty} \gamma^{h-1} r_h \mid \pi, \tau]$ in $\tau$.

**Supervised Pretraining.** Our approach builds on the DPT architecture [16], a supervised pretraining method designed to equip transformer models with ICRL capabilities. In DPT, a set of tasks $\{\tau^i\}_{i=1}^m$ is drawn from a task distribution $p_\tau$. Each task $\tau^i \in \mathcal{M}$ represents an instance of an MDP where $\mathcal{M}$ is the space of all tasks of interest, and for each task, a context dataset $D^i$ is generated from interactions between a behavioral policy and $\tau^i$, i.e., $D^i = \{(s_h^i, a_h^i, s_{h+1}^i, r_h^i)\}_h$, where $a_h^i$ is selected by the behavioral policy. For each task $\tau^i$, a query state $s_{\text{query}}^i \in \mathcal{S}$ is chosen, and the optimal action $a_i^\star$ is sampled from $\pi_{\tau^i}^\star(s_{\text{query}}^i)$, where $\pi_{\tau^i}^\star$ is the optimal policy for $\tau^i$. The complete pretraining dataset is then denoted as $\mathcal{D}_{pre} = \{D^i, s_{\text{query}}^i, a_i^\star\}_{i=1}^m$. Let $T_\theta$ represent a causal GPT-2 transformer with parameters $\theta$ [28]. The pretraining objective of DPT is formulated as follows:

$$\min_\theta \frac{1}{m} \sum_{i=1}^m -\log T_\theta\left(a_i^\star | s_{\text{query}}^i, D^i\right). \tag{1}$$

**ICRL Deployment.** The pretrained autoregressive TM $T_\theta$ can be deployed as an agent in both **offline** and **online** settings. During deployment, an unseen task $\tau$ is sampled from the task distribution $p_\tau$. In offline deployment, a dataset $D_{\text{off}}$ is first collected from $\tau$, typically using trajectories generated by a random policy. Once available, DPT selects actions based on the policy $T_\theta(\cdot|s_h, D_{\text{off}})$ after observing the state $s_h$ at time step $h$. For online deployment, DPT begins with an empty dataset $D_{\text{on}}$. At each episode, DPT follows $T_\theta(\cdot|s_h, D_{\text{on}})$ to collect a trajectory $\xi = \{s_1, a_1, r_1, \ldots, s_H, a_H, r_H\}$, which is appended to $D_{\text{on}}$. This process repeats for a predefined number of episodes. The pseudocode for this process is provided in Algorithm 1 in Appendix H.

## 3 Extending Robust RL for Robust ICRL

To improve the robustness of ICRL methods, a natural starting point is to adapt existing robust RL techniques developed for standard RL settings to the ICRL context. In Section 3.1, we briefly review **adversarial training**, a widely used technique for enhancing policy robustness in standard RL. Then, in Section 3.2, we explore how adversarial training can be directly extended to ICRL within the supervised pretraining framework and highlight its **inherent limitations** in this setting. *These limitations, in turn, motivate our proposed frameworks* in Section 4, which leverage the in-context learning capabilities of transformer models to overcome these challenges more effectively.

### 3.1 Robust RL From Adversarial Training

While the objective of standard RL is to find a policy $\pi_\tau$ that performs well for a specific target MDP $\tau$, robust RL aims to ensure performance even when the deployment MDP $\tau'$ differs from the pre-specified target $\tau$ [22].

Adversarial training is one of the most effective robust RL methods, which proposes to learn a policy robust to adversarial attacks and disturbances [26]. Adversarial training has shown great success in improving the policy robustness to both **(i)** environment mis-specification and **(ii)** external disturbances. It can be formulated as a **Markov Game**, defined by a tuple of 6 elements $\tau^G = (\mathcal{S}, \mathcal{A}, \mathcal{A}^a, P_\tau^G, R_\tau^G, \gamma, \rho_\tau)$; here, $\mathcal{S}$, $\gamma$, and $\rho_\tau$ are defined as in the MDP formulation, representing the set of states, the discount factor, and the initial state distribution, respectively; $\mathcal{A}$ and $\mathcal{A}^a$ are respectively the sets of actions that the agent (protagonist) and the adversary can take; $P_\tau^G : \mathcal{S} \times \mathcal{A} \times \mathcal{A}^a \to \Delta(\mathcal{S})$ is the transition function that describes the distribution of the next state given the current state and actions taken by the agent and the adversary; $R_\tau^G : \mathcal{S} \times \mathcal{A} \times \mathcal{A}^a \to \mathbb{R}$ is the reward function for the agent. Consider a parameterized adversary $\pi_\phi^a : \mathcal{S} \to \Delta(\mathcal{A}^a)$ where $\phi$ is its parameter. We use $\pi : \mathcal{S} \to \Delta(\mathcal{A})$ to denote the agent policy to learn. Let $s_h \in \mathcal{S}$ be the state of the environment, and let $a_h \in \mathcal{A}$ (respectively $a_h^a \in \mathcal{A}^a$) denote the action of the agent (respectively the adversary) at time step $h$. We use

$$\mathcal{R}_\tau(\pi, \phi) := \mathbb{E}\left[\sum_{t=0}^{\infty} \gamma^t R_\tau^G(s_h, a_h, a_h^a)|s_0 \sim \rho_\tau\right] \tag{2}$$

where $a_h \sim \pi(s_h), a_h^a \sim \pi_\phi^a(s_h)$ to represent the cumulative discounted reward that the agent $\pi$ can receive under the disturbance of the adversary following policy $\pi_\phi^a$. For a specific target MDP $\tau$, the **objective of adversarial training** for robustness is defined as

$$\pi_\tau^{rb} \in \underset{\pi}{\arg\max} \ \underset{\phi \in \Phi(\tau)}{\min} \ \mathcal{R}_\tau(\pi, \phi), \tag{3}$$

where $\Phi(\tau)$ is a pre-defined adversary space for $\tau$. In this approach, the RL agent $\pi$ optimizes the worst-case performance among all the adversarial disturbances from $\Phi$.

### 3.2 In-Context Robust Reinforcement Learning

Although primarily designed for standard RL scenarios, the adversarial training approach can be directly adapted to ICRL within the supervised pretraining framework. We term this approach *In-context Robust RL* (IC2RL), which we illustrate in Figure 1(b) and elaborate next.

Specifically, given the pretraining tasks $\{\tau^i\}_{i=1}^m$, we can construct a dataset for supervised pretraining following two steps. In the first step, we solve for a robust policy $\pi_{\tau^i}^{rb}$ for each task $\tau^i$, defined as

$$\pi_{\tau^i}^{rb} \in \underset{\pi}{\arg\max} \ \underset{\phi \in \Phi(\tau^i)}{\min} \ \mathcal{R}_{\tau^i}(\pi, \phi), \tag{4}$$

where $\Phi(\tau^i)$ is the adversary set for task $\tau^i$. The second step mirrors the data collection process of supervised pretraining in Section 2, except that the sampled query state $s_{\text{query}}^i \in \mathcal{S}$ is annotated with a **robust action label** $a_i^{rb}$ drawn from the robust policy $\pi_{\tau^i}^{rb}(s_{\text{query}}^i)$ rather than the optimal policy $\pi_{\tau^i}^*$ used in standard supervised pretraining. Thus, the complete pretraining dataset for IC2RL is $\mathcal{D}_{pre}^{rb} = \{D^i, s_{\text{query}}^i, a_i^{rb}\}_{i=1}^m$, and we can pretrain a causal transformer with the same pretraining objective as in (1). In this approach, the TMs are pretrained to infer an action which would be taken by a robust RL policy, conditioning on a given context (thus the name In-Context Robust RL). By definition in (4), the robust policies optimize their worst-case performance under disturbances. However, as a straightforward extension of the regular robust RL approaches, this approach inherits a couple of *long-standing limitations* within robust RL and induces *considerable computation cost*.

**First**, the max-min problem in (4) poses a challenging optimization problem, especially in the current era of deep RL, where such problems are often non-convex and non-concave. Solving this problem requires substantial computational effort to approximate robust RL for a single task. While this computational burden may be manageable in standard robust RL settings with only one task of interest, the (often vast) number of RL tasks involved in ICRL pretraining renders the required computational cost prohibitively high. **Second**, the success of the max-min approach in robust RL relies on effective construction of the adversary parameter set. If not properly constructed, an adversary set which is too broad leads to over-pessimistic policies due to the worst-case optimization, and, on the other hand, an adversary set which is too limited leads to policies with insufficient robustness to disturbances and environment mis-specification [6].

## 4 In-Context Adversarial Generalization and Adaptation

In Section 4.1, we present our framework, **In-Context Adversarial Generalization (ICAG)**, which overcomes the aforementioned limitations and effectively enhances the robustness of ICRL deployment. Recognizing ICAG may incur extra computational cost, in Section 4.2, we propose **In-Context Adversarial Adaptation (ICAA)** which bootstraps pretrained TMs to efficiently collect high-quality action labels with minimal data consumption.

### 4.1 In-Context Adversarial Generalization

**Motivational Insight.** The proposed framework is motivated by our insight that worst-case optimization is *not necessary* when the model can generalize in-context to various disturbances. In particular, instead of pretraining the TMs to infer actions taken by the robust policies that solve max-min problems, we propose to pretrain TMs to *directly infer the optimal actions to take under various disturbances*. Compared to regular ICRL, the proposed approach has two axes of generalization: one for generalizing across the environments and the other for generalizing across the disturbances. We next propose a supervised pretraining framework *In-Context Adversarial Generalization* (ICAG) to

achieve generalization on these two axes, which is illustrated in Figure 1(c). Notably, ICAG addresses the aforementioned challenges by entirely circumventing the max-min optimization.

**Algorithm.** Recall that $\mathcal{M}$ is a set of MDPs representing the space of environments (RL tasks) of interest for ICRL. Our goal is to pretrain TMs to generalize across the following *adversary-augmented* task space

$$\mathcal{M}_v := \{(\tau, \phi) : \tau \in \mathcal{M}; \phi \in \Phi(\tau)\},$$

where $\Phi(\tau)$ is defined as in (4), representing a pre-specified set of adversaries for $\tau$. Here, each pair $(\tau, \phi) \in \mathcal{M}_v$ represents an environment $\tau$ along with an associated disturbance policy $\pi_\phi$. When an agent is deployed into an environment $\tau$ with a disturbance policy $\pi_\phi$, from the perspective of the agent, the disturbance $\pi_\phi$ is an integral part of the environment $\tau$. Thus, $\tau$ and $\phi$ jointly create an environment. We refer to such a new environment created by incorporating an adversary $\phi$ into an environment $\tau$ as a *variation environment*, denoted by $(\tau, \phi)$. Specifically, the MDP of this environment can be defined as $(\mathcal{S}, \mathcal{A}, P_{\tau,\phi}, R_{\tau,\phi}, \gamma, \rho_\tau)$, where $\mathcal{S}$, $\mathcal{A}$, $\gamma$, $\rho_\tau$ are defined as in the MDP formulation in Section 2, and $P_{\tau,\phi}(s, a) = \mathbb{E}_{a' \sim \pi_\phi(s)}[P_\tau^G(s, a, a')]$ and $R_{\tau,\phi}(s, a) = \mathbb{E}_{a' \sim \pi_\phi(s)}[R_\tau^G(s, a, a')]$ with $P_\tau^G$ and $R_\tau^G$ defined as in the Markov Game formulation in Section 3.1.

To pretrain TMs to generalize across $\mathcal{M}_v$, we sample a set of variation environments belonging to $\mathcal{M}_v$ for supervised pretraining. To this end, given a set of $m$ pretraining tasks $\{\tau^i\}_{i=1}^m \subset \mathcal{M}$, we create $K$ variations of every pretraining task $\tau^i$ by randomly sampling $K$ adversaries $\{\phi^{i,k}\}_{k=1}^K \subset \Phi(\tau^i)$ and including the sampled adversaries into $\tau^i$. Thus, we create a set of variation environments $\{(\tau^i, \phi^{i,k})\}_{i \in [m], k \in [K]}$. For every variation environment, we sample an in-context dataset $D$, a query state $s_{\text{query}}$, and an optimal action label following the optimal policy $\pi_{\tau,\phi}^\star(s_{\text{query}})$, defined as

$$\pi_{\tau,\phi}^\star \in \operatorname*{argmax}_\pi \mathcal{R}_\tau(\pi, \phi). \tag{5}$$

We note that the objective function in (5) does not involve max-min optimization. Moreover, learning optimal policies for $K$ variation environments can be easily parallelized. This can significantly improve the efficiency the pretraining. See Algorithm 2 in Appendix H for its pseudocode.

**ICAG Augments Tasks.** On top of improving the robustness of TMs for ICRL, ICAG can also be viewed as a data augmentation method. In particular, consider the adversary set $\Phi(\tau)$, which includes the null disturbance $\phi_0$ such that the MDP of the variation environment $(\tau, \phi_0)$ remains identical to the original MDP $\tau$. This can be trivially achieved by simply omitting an adversary when augmenting the pretraining tasks. In this case, the augmented task space $\mathcal{M}_v$ covers the original task space $\mathcal{M}$, i.e., $\mathcal{M} \subset \mathcal{M}_v$. Thus, ICAG effectively *increase the diversity and number of pretraining tasks*, leading to improved ICRL performance, as shown by the extensive empirical evidence in Section 5.

## 4.2 In-Context Adversarial Adaptation

The proposed ICAG framework has one notable limitation: obtaining optimal action labels for a variation environment $(\tau, \phi)$ is computationally intensive, as it requires training a policy *from scratch* to approximate the optimal policy $\pi_{\tau,\phi}^\star(s_{\text{query}})$ defined in (5). To address this challenge, we propose leveraging a pretrained ICRL model to efficiently generate high-quality, though not necessarily optimal, action labels that are sufficient to improve robustness through iterative supervised updates.

We introduce *In-Context Adversarial Adaptation* (ICAA), a scalable method that uses an existing ICRL model to efficiently adapt to each variation environment $(\tau^i, \phi^{i,k})$ and generate query state–action label pairs for pretraining. Given an initial dataset $\mathcal{D}^0$, we first train an initial model $T_\theta^0$ with supervised pretraining on $\mathcal{D}^0$ and then deploy it in each variation environment. Here, $\mathcal{D}^0$ can be the pretraining dataset containing only trajectories from the original, unaugmented environments. Starting with an empty context buffer, the model performs online deployment by interacting with the environment and incrementally updating its context using its own rollouts, yielding a trajectory sequence $\xi_0, \xi_1, \ldots, \xi_N$ (see Section 2 for details).

With these trajectories $\{\xi_n\}_{n=1}^N$, we construct a new pretraining dataset for each variation environment to update the current ICRL model $T_\theta^0$. Recall that an instance of the pretraining dataset $\{D, s_{\text{query}}, a^\star\}$ has three elements: a context dataset $D$ of trajectories, a query state $s_{\text{query}}$, and its corresponding action label $a^\star$. To this end, we first collect the initial $\underline{N} < N$ trajectories $\{\xi_n\}_{n=0}^{\underline{N}}$ to construct the context datasets for pretraining. Next, we independently sample state–action pairs $\{s_{\text{query}}^n, a_n\}_{n=0}^{\underline{N}}$

from the remaining trajectories $\{\xi_n\}_{n=\underline{N}}^N$. The sampled pairs are used as the query state-action label pairs for pretraining. In particular, as the performance of ICRL models increases with more trajectories during deployment, we sample from later trajectories $n > \underline{N}$ to focus on trajectories with higher cumulative returns and obtain action labels with higher quality.

This yields an environment-specific pretraining dataset of the form $\{D_n = \xi_n, s_{\text{query}}^n, a_n\}_{n=0}^{\underline{N}}$. Aggregating such datasets across all $m$ tasks and their respective $K$ variations, we combine them with $\mathcal{D}^0$ to construct an augmented training set $\mathcal{D}^1$. We then update the ICRL model by supervised pretraining on $\mathcal{D}^1$ to obtain $T_\theta^1$, which improves robustness and generalization *without requiring task-specific policies to be trained from scratch*. This process is repeated for $J$ rounds to iteratively refine the model $T_\theta^J$ on $\mathcal{D}^J$. See Algorithm 3 in Appendix H for a detailed procedure and the ablation study on the number of model refinement in Dark Room environment in Appendix D.

**Why ICAA Can Improve ICRL Performance.** While the convergent policies of ICRL models in variation environments do not necessarily have optimal performance, their action labels are still helpful in improving the ICRL models' performance, as we select out the improved actions generated by ICRL models for supervised pretraining. In other words, the actions generated by the ICRL models after convergence in online deployment have better performance than its original performance in the variation environments. Consequently, *the good actions get reinforced* during the sequential supervised finetuning, leading to improved ICRL performance.

## 4.3 Theoretical Analysis

We provide theoretical guarantees for both ICAG and ICAA to gain further insights into their efficacy. We present informal versions of our results to provide insights and defer our full results to Appendix E.

**Theorem (Informal) ICAG Generalization.** *Assume that the pretraining is sufficiently well, then ICAG pretrains TMs for implicit Posterior Sampling over both the deployment task and the potential adversary.*

In particular, Posterior Sampling (PS) is widely recognized as one of the most sample-efficient algorithms for a broad class of sequential decision-making problems [25]. By implicitly performing PS during deployment over *both* the deployment environment and potential adversarial perturbations, transformer models pretrained by ICAG are able to act optimally in the presence of an adversary capable of modifying the environment. This design enables ICAG to achieve *efficient robustness*, adapting to new tasks and perturbations with minimal sample complexity and without sacrificing performance for robustness.

**Theorem (Informal) ICAA Self-Improvement.** *Under mild generalization assumptions on the supervised pretraining framework, ICAA is equivalent to sequential supervised pretraining where action labels are generated by a policy whose performance does not degrade over the course of algorithm iterations.*

The generalization assumptions essentially require the DPT model to demonstrate better performance with context datasets collected by stronger behavioral policies. This has been well-established through extensive experiments in [16]. Under this assumption, ICAA can *progressively refine its action labels* through iterative self-labeling, leading to steady performance and robustness improvements until its performance plateaus. However, we emphasize that convergence to fully optimal action labels is generally unattainable without unrealistic conditions such as an unlimited number of exploratory trajectories. Thus, ICAA should be understood as an *data efficient complement to ICAG*, designed to quickly enhance model robustness with minimal data consumption, rather than to guarantee optimality under environment perturbations.

## 5 Experiments

We evaluate our proposed methods across three benchmarks compared against standard baselines. Specifically, in Dark Room [15] and Meta-World [46], we compare with in-context learning methods, including Decision Pretrained Transformer (DPT) [16], Mixed Decision Pretrained Transformer (mix-DPT) [39], Prompt-based Decision Transformer (prompt-DT) [43], and Adversarially Robust Decision Transformer (ARDT) [34]. In MuJoCo [36], we additionally compare witht SAC [8], RARL [26], and QARL [31], alongside the in-context baselines. Further, we show that ICAG

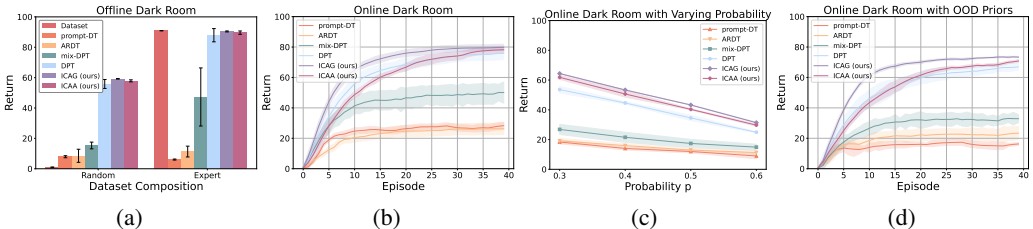

Figure 3: Performance on held-out Dark Room goals with average return over 10 random seeds. The error bar and the shaded area represent the standard error. (a) Offline evaluation given random and expert datasets. (b) Online evaluation without disturbances. (c) Online evaluation under disturbances with higher probability. (d) Online evaluation under disturbances with unseen priors.

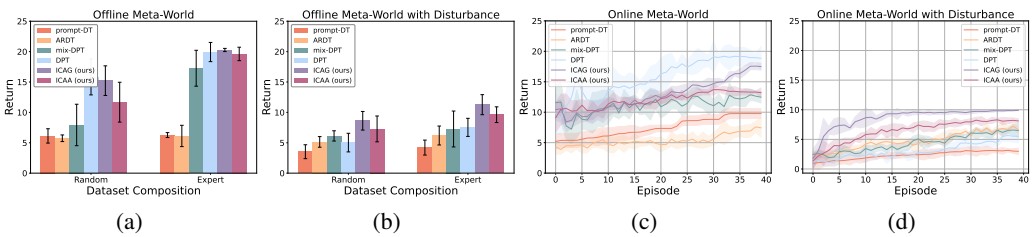

Figure 4: Performance on held-out Meta-World goals with average return over 10 random seeds. The error bar and the shaded area represent the standard error. (a) Offline evaluation given random and expert datasets. (b) Offline evaluation given random and expert datasets under disturbances. (c) Online evaluation without disturbances. (d) Online evaluation under disturbances.

generalizes better with environmental change in Appendix D. Full implementation and environment details are provided in Appendix B and Appendix C.

## 5.1 Dark Room

Dark Room is a sparse-reward navigation task in a $10 \times 10$ 2D discrete grid. For environment details and data generation, see Appendix C. Following the DPT evaluation protocol [16], we assess generalization to unseen tasks, by training on 80 goals and evaluating on 20 unseen goals. As shown in Figure 3(a), we report offline performance using both random and expert trajectories. All methods outperform the random trajectories (red), yet DT-based methods like prompt-DT and ARDT struggle with both random and expert test trajectories due to the randomness of the training set. In contrast, ICAG and ICAA outperform all baselines, including DPT, on both testing datasets, highlighting strong adaptation to expert-like behavior without direct exposure to expert trajectories.

In Figure 3(b) and Figure 3(d), we evaluate online adaptation over 40 episodes. Figure 3(d) introduces environmental disturbances with unseen priors, modeled using a Dirichlet distribution with takeover probability $p = 0.2$. ICAG achieves the highest return, while ICAA and DPT gradually converge toward ICAG's performance with increased adaptation. Under perturbations, ICAG's advantage becomes more pronounced in Figure 3(d). Finally, Figure 3(c) evaluates robustness across varying disturbance probabilities. The x-axis denotes the probability of adversarial action override, while the y-axis shows the average return after 40 episodes. ICAG consistently outperforms all other methods *under disturbance*, maintaining robustness even under high disturbance level (e.g., $p = 0.6$).

## 5.2 Meta-World

In the Meta-World benchmark, the agent is tasked with controlling a robotic hand to reach target positions in 3D space. Detailed descriptions of the environment and data generation process can be found in Appendix C. Meta-World includes 20 tasks in total. To assess the generalization capability of our approach to unseen reinforcement learning tasks, we train on 15 tasks and evaluate on the

remaining 5. Offline performance is evaluated using both random and expert trajectories, as shown in Figure 4(a) and Figure 4(b), while online performance is reported in Figure 4(c) and Figure 4(d). Across both offline and online settings, we observe that all methods exhibit performance degradation under state disturbances. Notably, while DPT suffers a significant drop in performance, ICAG and ICAA maintain higher robustness, consistently outperforming other methods under perturbation.

## 5.3 MuJoCo Control

We evaluate our methods on a diverse set of continuous control tasks from the DeepMind Control Suite [38], covering 6 environments and 11 tasks (see Appendix C). Pretraining datasets are built from historical trajectories of SAC agents. To improve robustness during ICAG supervised pretraining, we introduce $K$ fixed adversaries $\{\phi^{i,k}\}_{k=1}^{K}$ for each task $\tau^i$, each modeled by a separate neural network. SAC is trained to convergence under adversarial disturbances $\phi^{i,k}$, and the resulting trajectories form variation environments $\{(\tau^i, \phi^{i,k})\}_{i \in [m], k \in [K]}$ for $m$ pretraining tasks. Additional dataset details are in Appendix C.

We assess robustness from two perspectives: ($a$) adversarial disturbances affecting the agent, and ($b$) environment variations, such as changes in mass and friction. Under adversarial disturbances, ICAG demonstrates strong performance across Cartpole and locomotion tasks, particularly in high-dimensional control problems like Quadruped (see Figure 11 in Appendix D). Under environment variations, ICAG also outperforms other baselines in OOD settings, with the performance gap widening relative to in-distribution results. We present the full experimental results in Appendix D.

## 6 Related Works

Owing to space limitations, we only review the most relevant work here and defer the comprehensive literature review to Appendix A. We position our work at the intersection of offline RL, transformer-based decision-making, and robust RL. Offline RL typically focuses on learning policies from fixed datasets for the same tasks, using value pessimism or policy regularization to address distributional shift. In contrast, ICRL aims to generalize to unseen tasks without parameter updates by leveraging transformers pretrained on diverse trajectories. While recent transformer-based methods [3, 43, 16, 15] demonstrate promising generalization, they often remain vulnerable to perturbations and out-of-distribution shifts. Robust and adversarial RL [26, 33] enhance resilience by optimizing worst-case performance or introducing disturbances during training, but frequently incur high computational costs or degrade in-distribution performance. Our proposed methods, ICAG and ICAA, integrate the strengths of both paradigms to improve robustness and adaptability in ICRL.

## 7 Discussion and Conclusion

In this work, we propose two efficient frameworks—ICAG and ICAA—that enhance both robustness and generalization of transformer models for ICRL. ICAG improves robustness by augmenting pretraining tasks with perturbed environment variants, enabling the model to learn behaviors that generalize across potential disturbances. While this adds some computational cost, the overhead scales linearly with the number of variants and can be reduced via parallelized policy learning. To further improve data efficiency, we introduce ICAA, which lets a pretrained TM generate high-quality action labels for itself through interaction. This self-improvement loop quickly refines robustness using the model's generalization capability. Together, ICAG and ICAA provide complementary benefits: ICAG targets performance under adversarial or perturbed settings, while ICAA enables fast, data-efficient robustness enhancement. Practitioners can select the approach best suited to their computational and data constraints. We hope this work advances more robust and adaptable ICRL agents for real-world deployment.

### Acknowledgments

This work is sponsored in part by the AFOSR under the award number FA9550-19-1-0169, and by the NSF under NAIAD Award 2332744 as well as the National AI Institute for Edge Computing Leveraging Next Generation Wireless Networks, Grant CNS-2112562.

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

# A Related Work

**Offline RL.** Our work is situated within the broader field of offline RL [17, 21, 27]. Offline RL methods often employ approaches such as value pessimism or policy regularization to address the distributional shift between behavioral and optimal policies [41, 13, 14, 30, 45, 11, 4, 7]. While offline RL aims to solve the *same* tasks from which the offline datasets are collected, ICRL aims to efficiently generalize and adapt to *unseen* tasks.

**Transformers for Decision Making.** Autoregressive models [28, 1, 40, 37, 24] have achieved remarkable successes across various domains. Their application to sequential decision-making tasks, such as bandit and MDP problems, has shown transformers outperforming traditional methods [18, 47]. Decision Transformer (DT) [3, 49, 20, 44] formulates offline RL as return-conditioned supervised learning and scales well across multi-task settings. Methods like prompt-DT [43], Algorithm Distillation (AD) [15], and DPT [16] aim to enhance in-context generalization to new goals and tasks. Recent efforts [5, 50] further relax the assumptions on pretraining datasets by leveraging suboptimal trajectories or policies. However, challenges remain in addressing out-of-distribution (OOD) contexts with different environment dynamics and robustness to disturbances. To tackle these issues, we propose a transformer-based approach that leverages in-context learning to address the robustness problem and further utilize online adaptation to learn high-quality action without training separate policies for each adversarial task from scratch.

**Robust RL and Adversarial Training.** Robust reinforcement learning (RL) focuses on generalizing to out-of-distribution (OOD) environments by optimizing worst-case performance across various transition models [23, 10]. Deep RL methods commonly achieve robustness during training through parametric uncertainty, which considers a range of simulation parameters to optimize for worst-case performance [33, 29], or through adversarial training, which introduces perturbations to the environment (e.g., actions, observations, or transitions) to simulate potential deployment-time disturbances [26, 35, 48, 31, 9]. These methods enhance generalization to dynamics unseen during training but often sacrifice in-distribution (ID) performance.

# B Implementation Details

## B.1 Algorithm.

**Soft Actor-Critic (SAC) [8].** Soft Actor-Critic (SAC) is an off-policy reinforcement learning algorithm designed to balance exploration and exploitation by optimizing the trade-off between expected rewards and action entropy. SAC aims to learn a policy that not only maximizes long-term rewards but also encourages exploration by maximizing the entropy of the policy's action distribution. The algorithm utilizes an actor network to select actions, along with two critic networks that estimate the $Q$-values of state-action pairs. The learning objective is to maximize a soft Bellman equation: $J(\pi) = \sum_h \mathbb{E}_{(s_h, a_h) \sim D}[Q(s_h, a_h) - \alpha \log \pi(a_h|s_h)]$, where $Q(s_h, a_h)$ represents the value of the state-action pair as estimated by the critics, $\alpha$ is a temperature parameter controlling the exploration-exploitation balance, and $\pi(a_h|s_h)$ is the probability of selecting action $a_h$ in state $s_h$. SAC is trained by sampling mini-batches from a replay buffer to update both the policy and the $Q$-value estimates. We use the implementation without adversarial training from Reddi et al. [31].

**Decision-Pretrained Transformer (DPT) [16].** The Decision-Pretrained Transformer (DPT) is designed for in-context learning in reinforcement learning (RL) tasks by leveraging supervised pretraining. The key idea behind DPT is to train a transformer model to predict optimal actions for a given query state, using an in-context dataset that includes interactions from a variety of tasks. These interactions are represented as state-action-reward tuples, which provide the necessary context for decision-making. During pretraining, DPT samples a distribution of tasks. For each task $\tau_i$, an in-context dataset $D^i$, is constructed, consisting of sequences of state-action-reward transitions that reflect prior experience with that task. A query state $s_{query}^i$ is then sampled from the MDP's state space, and the model is trained to predict the optimal action based on both the query state and the task-specific context $D^i$. The training objective is to minimize the expected loss over the sampled task distribution, where the model learns to predict a distribution of actions given the query state and context. DPT shares the same set of finite environments as ICAG and ICAA, but its pretraining dataset does not consider robustness.

**Mixed Decision-Pretrained Transformer (mix-DPT) [39].** The mix-DPT framework extends DPT by splitting the learning process into two phases: the early training phase and the mixed training phase, addressing the out-of-distribution (OOD) issue between training and testing. During the early training phase, data are generated using a pre-specified decision function $f$, such as a random policy for the Dark Room task or SAC-trained policies for Meta-World and MuJoCo control. In the mixed training phase, data are generated using both the function $f$ and the current DPT model, with the proportion controlled by a hyper-parameter $\kappa$. In the experiments conducted by Wang et al. [39], historical trajectories include the optimal actions for each time step. To ensure a fair comparison, we follow their setup but use only a query state and the corresponding optimal action, as done in DPT and ICAG. For our experiments, we follow the parameter choice $\kappa = 1/3$ from Wang et al. [39], with the total number of training iterations set to 40% of the overall training iterations.

**Prompt-based Decision Transformer (prompt-DT) [43].** Prompt-DT builds upon Decision Transformer [3] and organizes its data to enable few-shot policy generalization through the use of trajectory prompts. For each task $T_i$, a prompt $\tau_i^*$ of length $K^*$ is constructed from a set of few-shot demonstrations $P_i$, which consist of state-action-reward-to-go tuples $(s^*, a^*, \hat{G}^*)$. This prompt captures the task-specific context required for policy adaptation. To further enrich the context, the most recent trajectory history $\tau_i$, sampled from an offline dataset $D_i$, is appended to the task-specific prompt. This forms the complete input sequence $\tau_{input}$ input. Specifically, the input sequence is represented as: $\tau_{input} = (\tau_i^*, \tau_i)$. This sequence consists of $3(K^* + K)$ tockens, following the state-action-reward format. The full sequence $\tau_{input}$ is then processed by a Transformer model, which autoregressively predicts the next actions corresponding to each state token. We follow the original prompt-DT setup and set $k = 20$. Prompt-DT uses the same pretraining dataset as DPT, but lacks query state-action pairs, highlighting the architecture's effectiveness in DPT-based methods (e.g., DPT, ICAG, ICAA).

**Adversarially Robust Decision Transformer (ARDT) [34].** ARDT enhances the Decision Transformer by associating worst-case returns-to-go via minimax expectile regression with trajectories to improve robustness against adversarial perturbations. Specifically, the estimated $Q$ values from expectile regression replaces the returns-to-go in vanilla DT during training.

**Robust Adversarial Reinforcement Learning (RARL) [26].** RARL trains a protagonist to compete against destabilizing forces introduced by an adversary in a zero-sum Markov game, where the optimal strategy (i.e., the rational strategy) corresponds to a Nash equilibrium. In this setup, the protagonist selects actions to maximize performance, while the adversary is trained to take actions that minimize the same performance metric. By training under these destabilizing perturbations, the protagonist learns to develop robust skills that help it counter distribution shifts and adversarial attacks when deployed in real-world scenarios. We implement RARL using the framework from Reddi et al. [31], where both the protagonist and adversary are represented as agents with SAC policies.

**Quantal Adversarial Reinforcement Learning (QARL) [31].** QARL formulates a robust adversarial reinforcement learning objective with entropy regularization, designed to model Markov games under bounded rationality. It introduces two temperature coefficients for the Shannon entropy of both the protagonist's and adversary's action distributions, allowing the optimization problem between the two players to be framed as a Quantal Response Equilibrium (QRE). QRE is a generalization of the Nash equilibrium, extending it to scenarios where agents may not act with complete rationality. We implement QARL using the framework from Reddi et al. [31], where both the protagonist and adversary are represented by SAC-based agents.

### B.2 Model Architecture and Hyper-parameters

**Transformer.** Our models, DPT, MDPT, and prompt-DT are all based on a causal GPT-2 architecture [28]. It consists of 4 attention layers, each with a single attention head. In DarkRoom, the embedding size is 32, while Meta-World and MuJoCo environments' embedding size are 256. Prompt-DT and ARDT are built on Decision Transformer, separating the individual $(s, a, s', r)$ into their own embeddings to be made into one long sequence. The remaining transformer-based baselines and our models view the transition tuples in the dataset as their own singletons, to be related with other singletons in the dataset through the attention mechanism. We use the AdamW optimizer with a weight decay of $1e - 4$, a learning rate of $1e - 3$, and a batch size of 128.

**Multilayer perceptron (MLP).** For all non-transformer agents, e.g., SAC, RARL, QARL, we directly list their shared hyper-parameters and architecture in Table 1.

Table 1: Non-transformer agent hyper-parameters (e.g, SAC, RARL, QARL)

| Hyper-parameters | Values |
|---|---|
| No of hidden layers | 3 |
| No of hidden units per layer | 256 |
| activation function | ReLU |
| optimizer (actor and critic) | Adam |
| actor learning rate | $1 \cdot 10^{-4}$ |
| critic learning rate | $3 \cdot 10^{-4}$ |
| initial replay memory size | $3 \cdot 10^{3}$ |
| max replay memory size | $1 \cdot 10^{6}$ |
| warmup transitions | $5 \cdot 10^{3}$ |
| batch size | 256 |
| target smoothing coefficient | $5 \cdot 10^{-3}$ |
| initial temperature | $5 \cdot 10^{3}$ |
| temperature learning rate | $3 \cdot 10^{-4}$ |

## C  Environment Settings and Pretraining Dataset

### C.1  Darkroom.

Dark Room is a sparse-reward navigation task set in a discrete $10 \times 10$ grid. At the beginning of each episode, the agent is randomly placed in one of the grid cells, while the goal location is hidden and fixed at a random cell. The agent receives an observation of its current $(x, y)$ position and selects from five discrete actions: move left, right, up, down, or stay in place. The episode horizon is $H = 100$ steps. The agent receives a reward of $r = 1$ only when it reaches the goal, and $r = 0$ otherwise. At test time, the agent always starts from the origin $(0, 0)$.

To construct the pretraining dataset, we generate $100,000$ trajectories using a uniform-random policy, evenly distributed across $100$ different goal locations. For each, we sample query states uniformly and compute optimal actions using a heuristic that first aligns the agent's $y$-coordinate with the goal and then the $x$-coordinate. The dataset is partitioned into $80,000$ training examples (corresponding to $80$ goal locations) and $20,000$ validation examples (corresponding to the remaining $20$ goals).

To improve robustness, we augment the environment with adversarial perturbations. Specifically, for each of the $m$ training tasks $\{\tau^i\}_{i=1}^m$, we construct a set of perturbed environments $\mathcal{M}_v = \{(\tau^i, \phi^{i,k})\}_{i \in [m], k \in [K]}$ by allowing an adversarial action to override the agent's intended action with probability $p \sim \mathcal{U}(0.0, 0.2)$. The adversarial action is sampled from a Dirichlet distribution by priors $\boldsymbol{\alpha}$, with the most probable action selected. With probability $1 - p$, the agent executes its original action. This construction models each perturbed task as a probabilistic MDP [35], capturing structured uncertainty in action dynamics.

The Dirichlet distribution used satisfies $\sum_{i=1}^{I} x_i = 1$ and $x_i \in [0, 1]$ for all $i$. The probability density function is given by:

$$f(x_1, ..., x_I; \alpha_1, ..., \alpha_I) = \frac{1}{\beta(\boldsymbol{\alpha})} \prod_{i=1}^{I} x_i^{\alpha_i - 1}.$$

, where $\beta(\boldsymbol{\alpha})$ is the multivariate beta function, which can be expressed in terms of the gamma function

$$\beta(\boldsymbol{\alpha}) = \frac{\prod_{i=1}^{I} \Gamma(\alpha_i)}{\Gamma(\sum_{i=1}^{I} \alpha_i)}, \qquad \boldsymbol{\alpha} = (\alpha_1, ..., \alpha_I).$$

These perturbed environments are used to generate additional trajectories in the style of DPT, increasing the diversity and robustness of the pretraining dataset.

## C.2 Meta-World.

We focus on the ML1 pick-place benchmark, where the agent must grasp an object and place it at a designated target location. Each task is defined by a 39-dimensional state space that includes the gripper's position and binary open/close state, the 3D pose of the object, and the coordinates of the target. The agent operates in a continuous action space, allowing it to adjust its end-effector position in three dimensions and control the gripper's open/close state to facilitate object manipulation.

The environment provides shaped rewards to guide learning, including incentives for approaching the object, establishing a grasp, transporting the object, and releasing it at the target. The goal is to learn a policy that effectively sequences these skills to solve the overall manipulation task. For generalization evaluation, we train on 15 task configurations and test on 5 held-out tasks with novel object and target positions.

To construct the pretraining dataset, we collect historical trajectories from agents trained using Soft Actor-Critic (SAC). For each task, SAC is trained until convergence, and we sample from the resulting trajectories to form offline datasets. The built-in deterministic policy is treated as the optimal expert policy.

To simulate real-world deployment challenges, we introduce robustness through adversarial perturbations to the observation space. Specifically, for each task $\tau^i$, we construct $K$ perturbed variations $\{(\tau^i, \phi^{i,k})\}_{i \in [m], k \in [K]}$, where each $\phi^{i,k}$ is implemented as a fixed-weight neural network with bounded magnitude ($|\phi^{i,k}| \leq 1$). These perturbations target only the perceptually estimated components of the observation—namely, object pose, end-effector position, and goal location—while avoiding internal robot states such as joint angles or velocities, which are typically less affected by real-world noise. This design mimics sensor uncertainty (e.g., from depth cameras or visual drift) without destabilizing control signals that are precise on physical hardware.

## C.3 MuJoCo.

The MuJoCo environments [36] used in our experiments are standard implementations from the DeepMind Control Suite [38]. Specifically, we consider modified versions of these environments that incorporate adversarial settings (i.e., RARL, QARL, and pretraining for ICAG). In each environment, adversarial actions and force magnitudes are carefully selected to challenge the agent and promote robust behavior. The adversarial action spaces are intentionally designed to differ from those of the protagonist agent to exploit domain knowledge. The adversary forces are calibrated to be sufficiently large to foster agent robustness and generalization, while still posing a significant challenge to the protagonist.

Table 2: MuJoCo Environment-specific parameters for adversarial training and (protagonist) agent's standard observation/action space

| Environment | Adversary max force | Adversary action space | Observation space | Action space |
|---|---|---|---|---|
| Cartpole | 0.005 | 2D forces on pole (1) | 5 | 1 |
| Cheetah | 1.0 | 2D force on feet & torso (6) | 17 | 6 |
| Hopper | 1.0 | 2D force on feet & torso (4) | 15 | 4 |
| Quadruped | 10 | 3D force on torso (1) | 78 | 12 |
| Reacher | 0.1 | 2D force on arm (2) | 6 | 2 |
| Walker | 1.0 | 2D force on feet (4) | 24 | 6 |

We conduct experiments in 6 environments shown in Figure 5. Then environment-specific parameters, along with the corresponding observation and action spaces for the standard environments, are detailed in Table 2. For all environments, the discount factor is set to 0.9 and the horizon is set to 200.

Each environment is associated with one or more problems, defined as instances of the model with specific Markov Decision Process (MDP) structures. For example, in the CartPole environment, we define two problems: **swingup**, where the pole starts pointing downward, and **balance**, where the pole begins near the upright position. The goal in both problems is to manipulate the forces applied to a cart at the base in order to either swing up or balance an unactuated pole. In the Reacher environment, a two-link planar arm must reach a randomized target location, with a reward of 1 when the end effector reaches the target sphere. Two problems are defined here: in the **easy** problem, the target sphere is larger than in the **hard** problem.

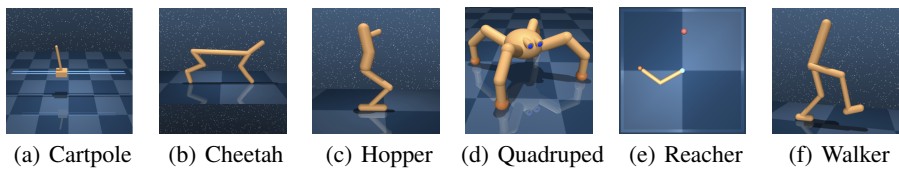

(a) Cartpole     (b) Cheetah     (c) Hopper     (d) Quadruped     (e) Reacher     (f) Walker

Figure 5: Illustrations of the MuJoCo environments.

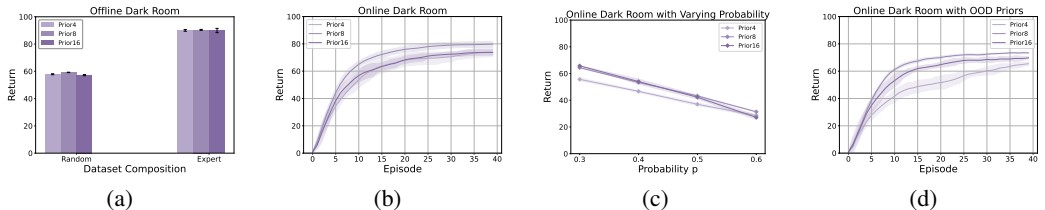

Figure 6: Ablation study for ICAG with different number of training priors on held-out Dark Room goals from test tasks with average return over $10$ random seeds. The error bar and the shaded area represent the standard error. (a) Offline evaluation given random and expert datasets. (b) Online evaluation without disturbances. (c) Online evaluation under disturbances with higher probability. (d) Online evaluation under disturbances with unseen priors. Note that prior $K \in \{4, 8, 16\}$ in each subfigure implies $K$ adversaries involved in pretraining dataset.

In the Hopper environment, a planar one-legged hopper is initialized in a random configuration and is rewarded based on torso height and forward velocity. The remaining three environments: Walker, Cheetah, and Quadruped focus on maximizing forward velocity. In the Cheetah environment, the reward is linearly proportional to the forward velocity, capped at a maximum of $10m/s$. The Walker environment includes two tasks: **walk** and **run**, which differ in their velocity requirements and include components to encourage upright posture and minimal lateral movement. For standard in-context reinforcement learning (ICRL) methods, such as DPT [16], We use $m$ pretraining tasks $\{\tau^i\}_{i=1}^m \subset \mathcal{M}$ with varying internal conditions. For each task $\tau^i$, we construct a pretraining dataset using 6000 historical trajectories collected from SAC agents trained to convergence.

We generate $K$ variations of each pretraining task $\tau^i$, incorporating adversaries $\phi^{i,k}$ to form a set of variation environments $\{(\tau^i, \phi^{i,k})\}_{i\in[m], k\in[K]}$. In our experiments, $m = 16$ tasks (from a $4 \times 4$ grid) and $K = 10$ variations, with hyper-parameter tuning in Appendix D.3.

We follow the SAC training approach in Reddi et al. [31], training $K = 10$ SAC policies for each $\tau^i$ under different variations, which are treated as distinct tasks. Each variation is initialized as a fixed-weight neural network ($|\phi^{i,k}| \leq 1$), unlike RARL or QARL, where adversarial policies are updated during training. These variations act as adversaries, with SAC policies trained under disturbance.

## D    Additional Experimental Results

### D.1    Ablation on number of priors for adversaries in Dark Room

We conduct ablation studies to assess the impact of varying the number of adversaries $\{\phi^{i,k}\}_{k=1}^K$ with different Dirichlet priors in the pretraining dataset to ICAG. The total number of pretraining trajectories remains consistent across different priors. Specifically, we explore $K \in \{4, 8, 16\}$ with varying $\alpha$ values (see Appendix C.1), and present the results in Figure 6.

In offline evaluation (shown in Figure 6(a)), both random and expert trajectories yield similar performance across different $K$ values. For online evaluation, Figure 6(b) shows results without disturbances, while Figure 6(d) introduces disturbances with a probability of $p = 0.2$, which were absent during pretraining. When fewer adversaries are used (e.g., $K = 4$), the agent requires more episodes to adapt to environments with unseen priors, as indicated in Figure 6(d).

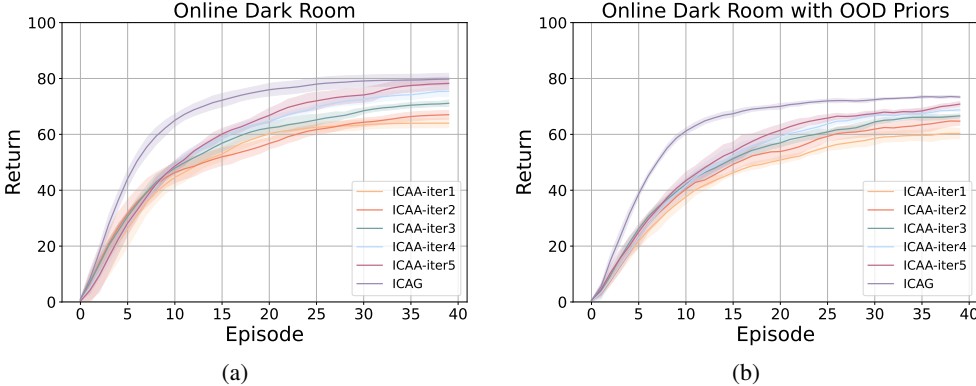

Figure 7: Performance on held-out Dark Room goals with average return over 10 random seeds among our methods. Note that ICAA-iter1 denotes the online evaluation with $T_\theta^1$. The error bar and the shaded area represent the standard error. (a) Online evaluation without disturbances. (b) Online evaluation under disturbances with unseen priors.

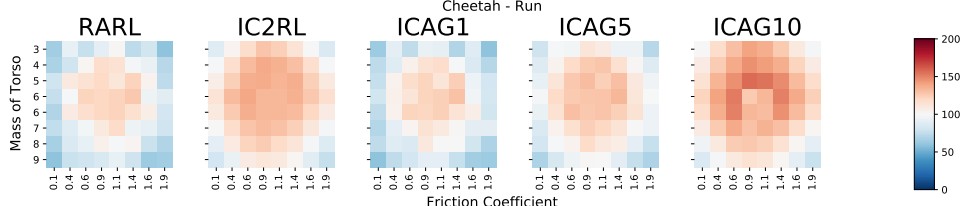

Figure 8: Generalization capability mainly among IC2RL and ICAG with varying $K$ variations, where ICAG $K$, with $K \in \{1, 5, 10\}$.

Finally, we evaluate all methods with a single adversary using a uniform distribution for disturbances, varying the disturbance probability in Figure 6(c). Based on these results, we select $K = 8$ for a better balance between online and offline performance, and use it for comparisons with other baselines in Figure 3.

### D.2 Ablation on the Number of ICAA Rounds in Dark Room

In Section 4.2 and Algorithm 3, ICAA employs an online approach to generate high-quality action labels and iteratively refine the model $T_\theta^J$ using the dataset $\mathcal{D}^J$, thereby reducing dependence on optimal action labels. We present the online evaluation results both with and without disturbances in Figure 7, and compare them against ICAG. Here, ICAA-iter1 represents the online evaluation using the model $T_\theta^1$. The results demonstrate that increasing the number of refinement iterations consistently improves performance under both without and with disturbances. However, the performance of ICAA remains bounded above by ICAG, as ICAG directly leverages optimal action labels.

### D.3 Variant of ICAG and IC2RL

As discussed in Section 3.2, extending traditional robust adversarial RL approaches to in-context settings presents challenges. To explore this, we conduct experiments in the Cheetah environment, where for each task $\tau^i$ with 16 pairs of internal conditions, we learn a robust policy via RARL, as shown in Figure 1(b). In Figure 8, we demonstrate that IC2RL improves RARL performance, but it is relatively impractical, solving max-min optimizations for all tasks. Additionally, we investigate the effect of varying the number of adversaries $K$ in $\{(\tau^i, \phi^{i,k})\}_{i\in[m],k\in[K]}$. We show that increasing $K$ acts as a data augmentation technique, supporting our statement in Section 4.1 that the augmented task space $\mathcal{M}_v$ covers the original task space $\mathcal{M}$ ($\mathcal{M} \subset \mathcal{M}_v$), leading to better generalization. Notably, ICAG10 with $K = 10$, as shown in Figure 8, outperforms IC2RL.

### D.4 MuJoCo Performance and Robustness

Notably, we use SAC agents in an adversarial training framework in both RARL and QARL, making SAC a natural baseline. Our adversarial setup follows [31], with environment-specific parameters detailed in Table 2. We investigate two types of robustness:

**Robustness to Agent Disturbance.** We measure this by evaluating learned policies under a worst-case adversary, which minimizes the agent's return without modifying the policy parameters.

**Robustness to Environmental Change.** We evaluate how well policies generalize to environments with variations in physical parameters such as mass and friction. Generalization is assessed by deploying policies in environments with shifted dynamics coefficients.

To analyze performance, we group MuJoCo tasks into three domains: cartpole, reacher, and locomotion (e.g., quadruped, cheetah, walker, hopper). For all experiments, both standard RL baselines (SAC and QARL) and ICRL methods are deployed in the same training environments used by SAC, favoring the former. *This gives SAC and QARL significant advantages* and explains why all ICRL methods other than ours are outperformed by QARL while *ICAG consistently outperforms QARL and other ICRL methods in all evaluations*.

Performance metrics are visualized through box plots and heatmaps in Figure 9, Figure 10, Figure 11, Figure 12, Figure 13, and Figure 14. Note: PDT denotes prompt-DT [43], and MDPT refers to mix-DPT [39]. Specifically, ICAG improves robustness against adversaries compared to DPT. Consistent with findings in standard robust RL [26, 6], we observe that training for robustness can also improve performance in disturbance-free settings. ICAG particularly excels in high-dimensional control problems, such as the quadruped task (Figure 11).

Furthermore, ICAG demonstrates strong generalization under out-of-distribution (OOD) environmental changes. In Figure 12, Figure 13, and Figure 14, we evaluate how methods adapt when tested on parameter grids not seen during training. Each $8 \times 8$ heatmap varies two internal environment parameters, while the pretraining dataset only covers the central $4 \times 4$ grid. ICAG consistently achieves the best performance on these OOD environments, outperforming all baselines in tests such as Figure 14(b) and Figure 14(d).

## E  Theoretical Results

Here we provide theoretical guarantees for ICAG to gain further insights into their efficacy. Complete proofs of results in this section can be found in Appendix F.

We present two results. Our **first** result show that ICAG addresses the adversaries (disturbances) encountered during deployment in a manner similar to Posterior Sampling (PS) [25], which is widely recognized as the most sample-efficient algorithm for many sequential decision-making problems. Our **second** result shows that ICAA continuously improve the quality of its action labels so that we can improve the performance of ICRL models in each ICAA iteration.

To facilitate analysis, we consider a slightly modified supervised pretraining framework similar to Lee et al. [16] and Lin et al. [19] for ICAG where, for any variation environment $(\tau, \phi)$, the TMs also condition on a sequence $\zeta_h = (s_1, a_1^\star, s_2, \ldots, s_h, a_h^\star)$ where $s_{1:h}$ follows the distribution $p_s \in \Delta(\mathcal{S}^h)$ and $a_j^\star \sim \pi_{\tau,\phi}^\star(s_h)$ where $\pi_{\tau,\phi}^\star$ is the optimal policy for $(\tau, \phi)$ as defined in (5). Thus, the joint distribution over $(\tau, \phi, D, s_{\text{query}}, \zeta_h)$ for ICAG pretraining is

$$P(\tau, \phi, D, s_{\text{query}}, \zeta_h) = p_\tau(\tau) p_{\Phi(\tau)}(\phi) p_D(D; \tau, \phi)$$
$$p_{\text{query}}(s_{\text{query}}) p_s(s_{1:h}) \prod_{j=1}^h \pi_{\tau,\phi}^\star(a_j^\star | s_h), \tag{6}$$

where $p_\tau$ and $p_{\Phi(\tau)}$ are the sampling distributions of the environment and disturbance for ICAG, respectively; $p_D(D; \tau, \phi)$ is the distribution of context dataset $D$ given the variation environment $(\tau, \phi)$; $p_{\text{query}}$ is the distribution for sampling query states. Given the joint distribution in (6), posterior distributions such as $P(\tau, \phi | D)$ are well-defined.

Consider the following general PS process for a fixed task $\tau'$ and disturbance $\phi'$: initialize the posterior distribution as the ICRL pretraining distribution $p_{\tau,\phi}^{(1)} = p_\tau p_{\Phi(\tau)}$, and initialize an empty dataset $D$ to

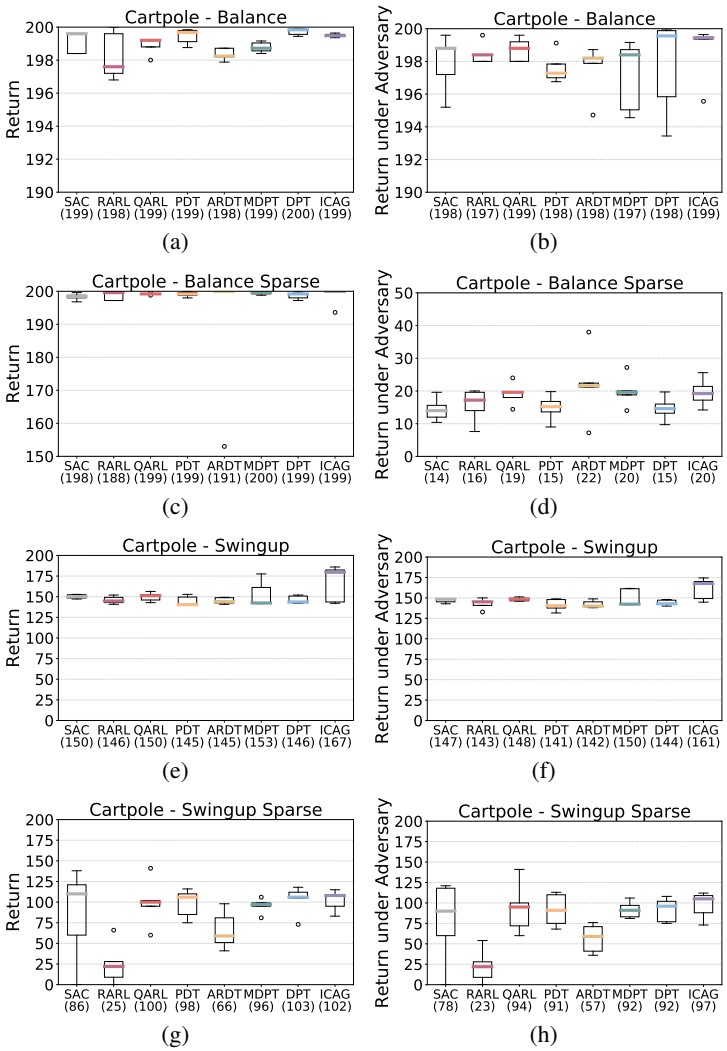

Figure 9: Performance and robustness on Cartpole problems (see the title of each boxplot), which are evaluated at the end of training without an adversary (left column) and against an adversary (right column). The number next to the name of each algorithm is the average performance across 10 seeds..

collect transitions; for $h \in \{1, \dots, H\}$: **(i)** sample a variation environment $(\tau^{(h)}, \phi^{(h)}) \sim p_{\tau,\phi}^{(h)}$; **(ii)** solve for $\pi_{\tau^{(h)},\phi^{(h)}}^{\star}$; **(iii)** given the current state $s^{(h)}$, take an action following $a^{(h)} \sim \pi_{\tau^{(h)},\phi^{(h)}}^{\star}(s^{(h)})$, and observe the reward $r^{(h)}$ and next state $s^{(h+1)}$; **(iv)** add the transition $(s^{(h)}, a^{(h)}, r^{(h)}, s^{(h+1)})$ into $D$, and update the posterior $p_{\tau,\phi}^{(h+1)} = P(\tau, \phi | D)$.

Note that although PS is provably sample-efficient, computing the posterior $P(\tau, \phi | D)$ is often intractable in practice. Next, *we prove that ICAG is an implicit PS*: during deployment, ICAG takes actions like the PS process above, *inferring the underlying environment and adversary without explicitly computing the posterior*. We first make some common mild assumptions for analysis [16].

**Assumption E.1.** Consider the context dataset $D = \{s_h, a_h, r_h, s_{h+1}\}_h$. The actions $a_h$ are conditionally independent of the variation environment $(\tau, \phi)$ given the history, i.e., $p_D(a_h | s_h, D_{h-1}) = p_D(a_h | s_h, D_{h-1}, \tau, \phi)$ where $D_h = \{s_{h'}, a_{h'}, r_{h'}, s_{h'+1}\}_{h' \leq h}$.

This assumption in essence assumes that the behavioral policies for collecting context datasets are functions of the history only. This holds, for example, when the context dataset is collected by random policies not depending on current states or any RL algorithms only using the history.

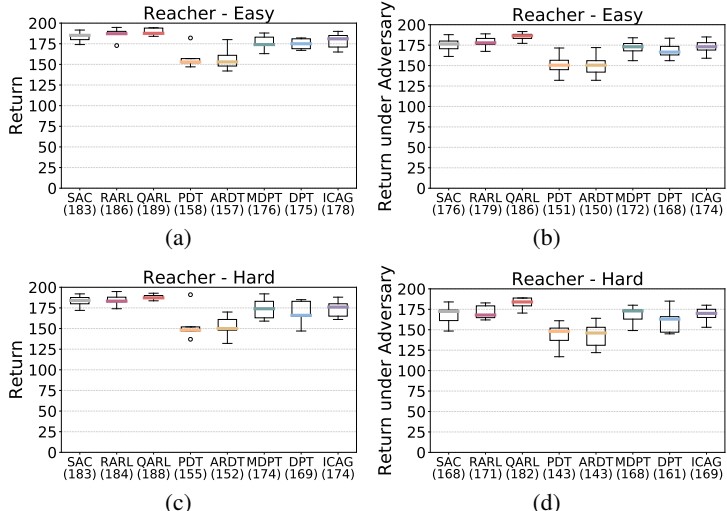

Figure 10: Performance and robustness on Reacher problems (see the title of each boxplot), which are evaluated at the end of training without an adversary (left column) and against an adversary (right column). The number next to the name of each algorithm is the average performance across 10 seeds.

**Assumption E.2.** Consider a sufficiently expressive and pretrained TM $T_\theta$. For all $(s_{\text{query}}, D, \zeta_h)$, $T_\theta(a|s_{\text{query}}, D, \zeta_h) = P(a|s_{\text{query}}, D, \zeta_h)$ for all $a$.

The purpose of Assumption E.2, which states that the pretrained TM $T_\theta$ matches the pretraining distribution $P$, is to focus the analysis on ICRL deployment rather than the quality of pretraining. This assumption is a common assumption for ICRL [16] and in-context learning analysis [42]. To see why this assumption is valid, it is well-established that deep learning models, such as $T_\theta$, are universal approximators [32]. Moreover, the maximum likelihood (ML)-based pretraining loss for $T_\theta$ is equivalent to find a minimizer of the following expected Kullback–Leibler (KL) divergence

$$\mathbb{E}_{(s_{\text{query}}, D, \zeta_h) \sim P} \left[ \text{KL} \left( P(a|s_{\text{query}}, D, \zeta_h) \| T_\theta(a|s_{\text{query}}, D, \zeta_h) \right) \right],$$

where $\text{KL}(p,q) = \mathbb{E}_p[\log(p/q)]$ is the KL divergence between two distributions. Assuming sufficient expressiveness, the above divergence can be minimized at $T_\theta = P$. Thus, with extra coverage assumptions regarding $P$ and sufficient amount of pretraining data, Assumption E.2 can hold with $T_\theta$ and $P$ arbitrarily close to each other. However, we omit this proof as this would distract the focus of the analysis. Next, we present our main theoretical results.

**Theorem E.3.** *Fix an environment $\tau'$ with a disturbance $\phi'$ for deployment and a context dataset $D \sim p_D(D; \tau', \phi')$ for the pretrained TM $T_\theta$ to condition on for ICRL. Consider the random sequence $\Upsilon_h = (S^{(1)}, A^{(1)}, S^{(2)}, A^{(2)}, \dots, S^{(h)}, A^{(h)})$. It holds that*

$$P_{PS}(\Upsilon_h | \tau', \phi', D) = P_\theta(\Upsilon_h | \tau', \phi', D),$$

*where (i) $P_{PS}(\Upsilon_h | \tau', \phi', D)$ is the distribution of $\Upsilon_h$ following the PS algorithm: $(\tau_{ps}, \phi_{ps}) \sim P(\tau, \phi | D)$, $S^{(1)} \sim \rho_{\tau', \phi'}$, for all $h' \leq h$, $A^{(h')} \sim \pi^\star_{\tau_{ps}, \phi_{ps}}(S^{(h')})$, $S^{h'+1} \sim P_{\tau', \phi'}(S^{(h)}, A^{(h)})$; (ii) $P_\theta(\Upsilon_h | \tau', \phi', D)$ is the distribution of $\Upsilon_h$ following ICRL with $T_\theta$: $S^{(1)} \sim \rho_{\tau', \phi'}$, for all $h' \leq h$, $A^{(h')} \sim T_\theta(a|S^{(h')}, D, \Upsilon_{h'-1})$, and $S^{h'+1} \sim P_{\tau', \phi'}(S^{(h)}, A^{(h)})$.*

Theorem E.3 states that the trajectory distribution under ICRL with the pretrained TM $T_\theta$ is the same as the trajectory distribution under PS, establishing that ICAG pretrains TMs for implicit PS. In particular, ICAG implicitly estimates the posterior distribution of the environment $\tau$ and the adversary $\phi$ so that *ICAG can act optimally if there exists an adversary $\phi$ that can perturb the environment $\tau$*. Moreover, this process of estimating and adapting to a potential adversary is provably sample-efficient given the optimal sample efficiency of PS.

Next, we show that ICAA can *refine its action labels in an iterative manner*.

**Assumption E.4** (Posterior Consistency). *For any pair of underlying task and adversary $\tau^\star, \phi^\star$, as the context dataset $D$ contains more transitions, the posterior $P(\tau, \phi | D)$ concentrates toward the true task and adversary, i.e., for any neighbor $U$ of $(\tau^\star, \phi^\star)$, it holds that $P(U|D) \to 1$ as $|D| \to +\infty$.*

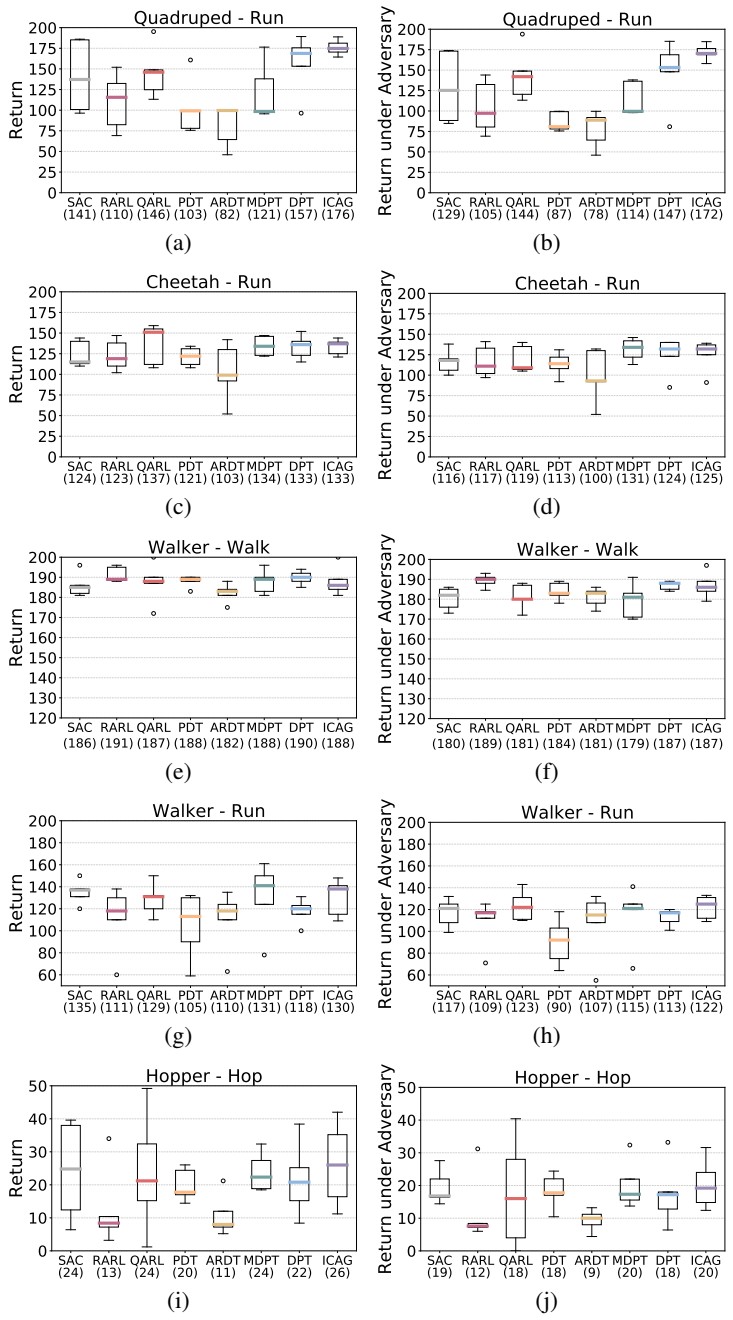

Figure 11: Performance and robustness on locomotion problems, i.e., Quadruped, Cheetah, Walker, and Hopper (see the title of each boxplot), which are evaluated at the end of training without an adversary (left column) and against an adversary (right column). The number next to the name of each algorithm is the average performance across 10 seeds.

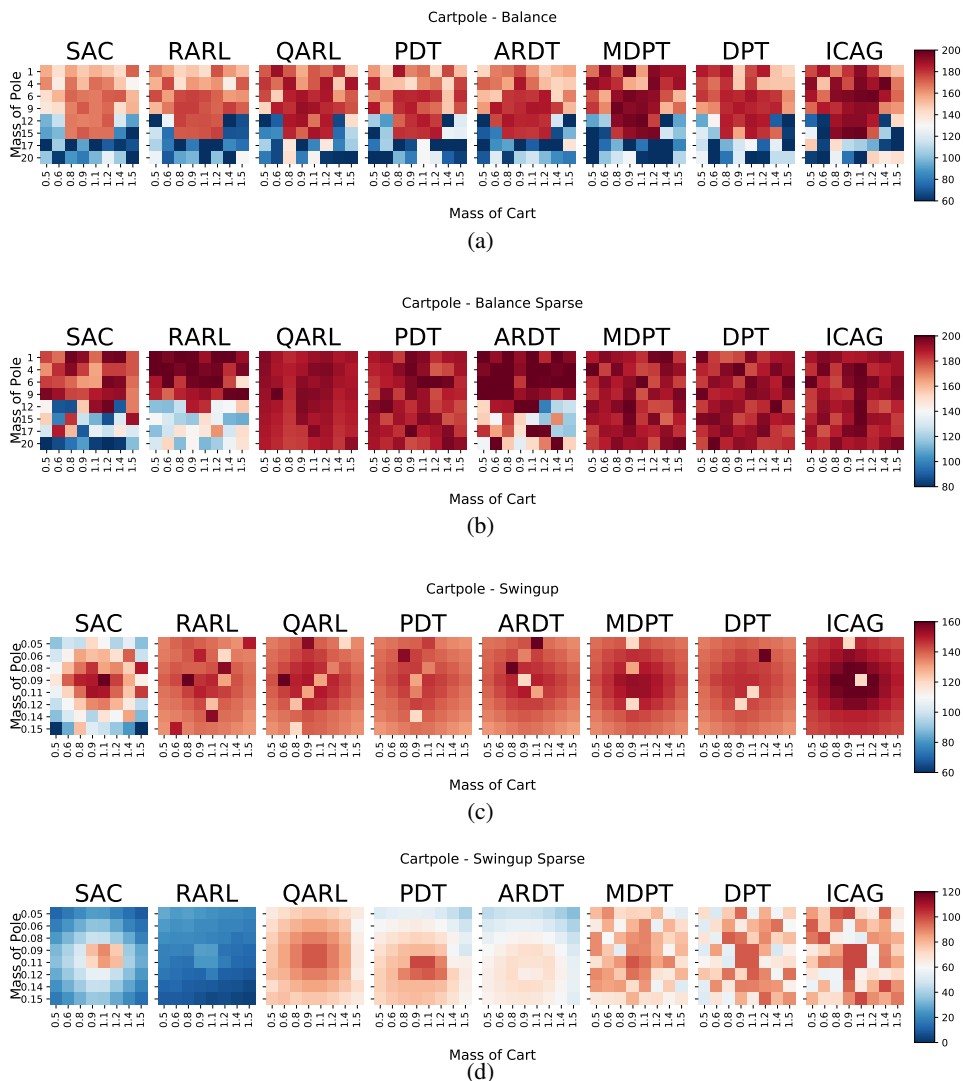

Figure 12: Robustness analysis on Cartpole (see the title of each heatmap set). Heatmaps show the performance obtained after training for varying properties of the environment, described in the $x - y$ axes. The number next to the name of each algorithm is the average performance across 10 seeds.

Assumption E.4 is a standard assumption and, in general, a fact for any Bayesian method. Following Theorem E.3 that ICRL models are performing Posterior Sampling, this implies that the transformer policy $T_\theta$ has increasing performance when the context size $|D|$ increases.

**Theorem E.5.** *Under Assumptions E.1, E.2 and E.4, in every iteration of ICAA (Algorithm 3), and for every variation environment $(\tau, \phi)$ with sufficient exploration, the action label generation policy achieves performance no worse than the transformer policy $T_\theta$ from the previous iteration within the same environment $(\tau, \phi)$.*

In particular, by matching the action labels generated by a stronger policy for the variation environment $(\tau, \phi)$ in the finetuning stage of each ICAA iterations, the performance of transformer policy $T_\theta$ increases and can generalize better to new tasks and adversaries. Theorem E.5 implies that *ICAA continues to improve the quality of its action labels until it saturates*.

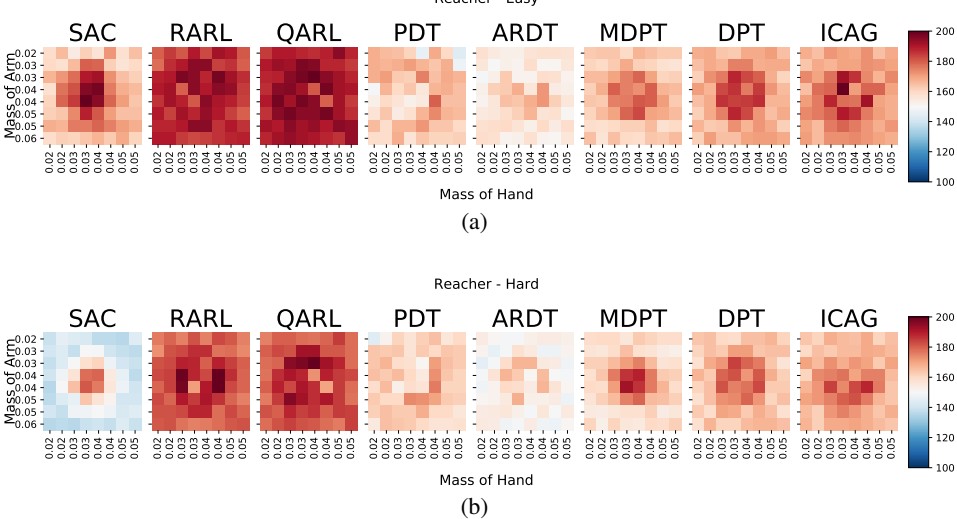

Figure 13: Robustness analysis on Reacher (see the title of each heatmap set). Each heatmap shows the performance obtained after training for varying properties of the environment, described in the $x - y$ axes. The number next to the name of each algorithm is the average performance across 10 seeds.

# F Proofs of Theoretical Results

## F.1 Proof of Theorem E.3

*Fix an environment $\tau'$ with a disturbance $\phi'$ for deployment and a context dataset $D \sim p_D(D; \tau', \phi')$ for the pretrained TM $T_\theta$ to condition on for ICRL. Consider the random sequence $\Upsilon_h = (S^{(1)}, A^{(1)}, S^{(2)}, A^{(2)}, \ldots, S^{(h)}, A^{(h)})$. It holds that*

$$P_{PS}(\Upsilon_h | \tau', \phi', D) = P_\theta(\Upsilon_h | \tau', \phi', D),$$

*where (i) $P_{PS}(\Upsilon_h | \tau', \phi', D)$ is the distribution of $\Upsilon_h$ following the PS algorithm: $(\tau_{ps}, \phi_{ps}) \sim P(\tau, \phi | D)$, $S^{(1)} \sim \rho_{\tau', \phi'}$, for all $h' \le h$, $A^{(h')} \sim \pi^\star_{\tau_{ps}, \phi_{ps}}(S^{(h')})$, $S^{h'+1} \sim P_{\tau', \phi'}(S^{(h)}, A^{(h)})$; (ii) $P_\theta(\Upsilon_h | \tau', \phi', D)$ is the distribution of $\Upsilon_h$ following ICRL with $T_\theta$: $S^{(1)} \sim \rho_{\tau', \phi'}$, for all $h' \le h$, $A^{(h')} \sim T_\theta(a | S^{(h')}, D, \Upsilon_{h'-1})$, and $S^{h'+1} \sim P_{\tau', \phi'}(S^{(h')}, A^{(h')})$.*

*Proof of Theorem E.3.* The proof is based on induction on $h$. Given that the results are for any fixed $\tau', \phi', D$, we omit the conditional dependence on them to improve clarity when there is no confusion. Recall that $P$ denotes the ICAG pretraining dataset distribution as defined in (6). We first prove a result to be used in the proof. Under Assumption E.1, we have

$$P_{PS}(\tau_{ps} = ps, \phi_{ps} = \phi | D) = P(\tau, \phi | D), \tag{7}$$

where $P(\tau | D)$ is the posterior distribution of the ICAG pretraining distribution $P$. This follows from

$$
\begin{aligned}
P_{PS}(\tau_{ps} = \tau, \phi_{ps} = \phi | D) &\propto P_{PS}(\tau_{ps}, \phi_{ps} = \phi, D) = P(\tau, \phi) P_{PS}(D | \tau, \phi) \\
&\propto P(\tau, \phi) \rho_{\tau', \phi'}(s_1) \prod_{(s_h, a_h, r_h, s_{h+1}) \in D} P_{\tau', \phi'}(s_{h+1} | s_h, a_h) p_D(a_h | D_{h-1}) \\
&\propto P(\tau, \phi) P(D; \tau; \phi) = P(\tau, \phi, D) \propto P(\tau, \phi | D),
\end{aligned}
$$

where the second $\propto$ is due to Assumption E.1 so that we can plug in $p_D(a_h | D_{h-1})$. Now we begin the proof of induction.

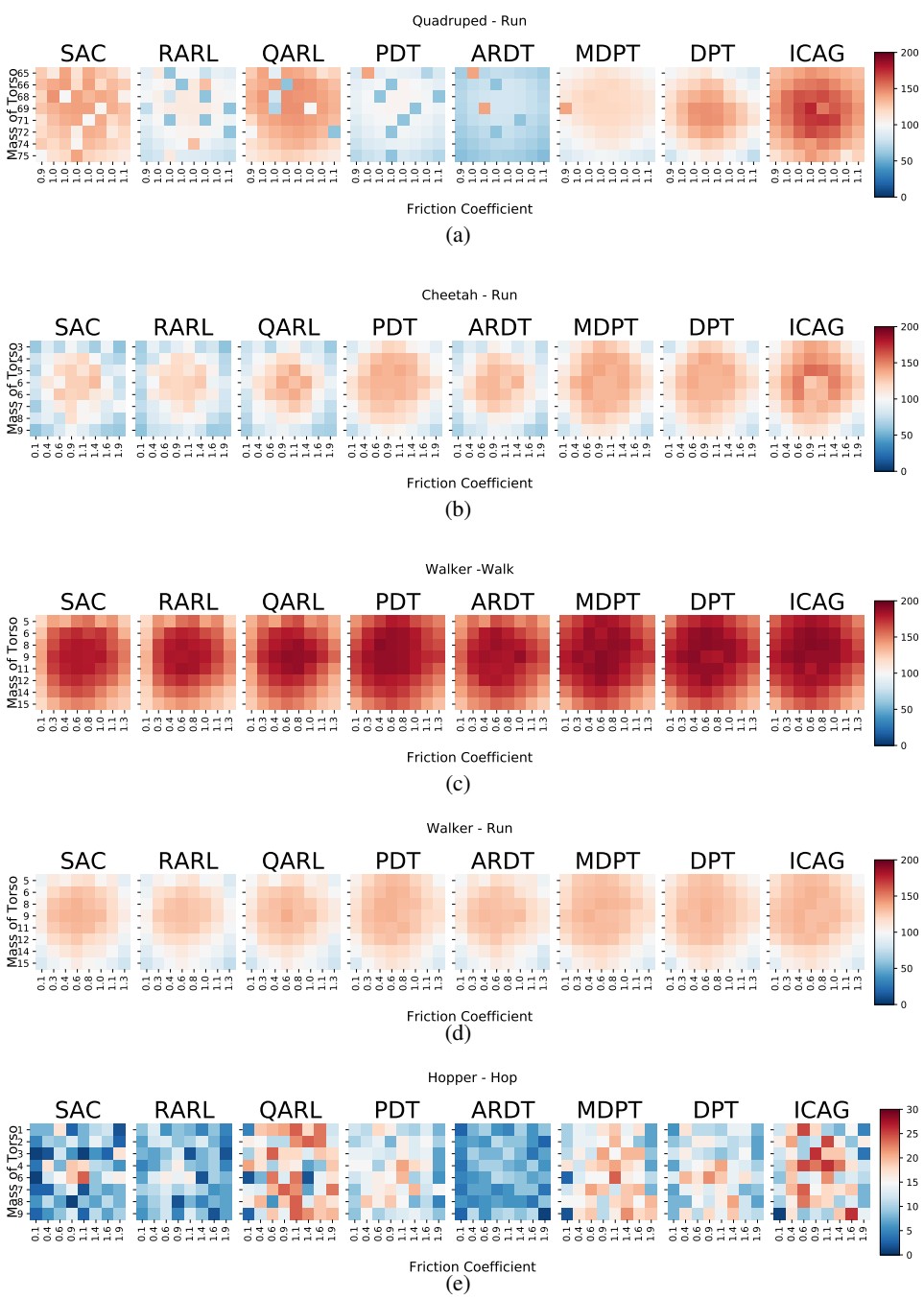

Figure 14: Robustness analysis on locomotion problems, i.e., Quadruped, Cheetah, Walker, and Hopper (see the title of each heatmap set). Each heatmap shows the performance obtained after training for varying properties of the environment, described in the $x - y$ axes. The number next to the name of each algorithm is the average performance across 10 seeds.

**Case $h = 1$.** We have

$$P_{PS}(S^{(1)}, A^{(1)}) = P_{PS}(S^{(1)})P_{PS}(A^{(1)}|S^{(1)}) = p_{\tau',\phi'}(S^{(1)}) \int_{\tau,\phi} P_{PS}(A^{(1)}, \tau_{ps} = \tau, \phi_{ps} = \phi|S^{(1)})d\tau d\phi$$

$$= p_{\tau',\phi'}(S^{(1)}) \int_{\tau} P_{PS}(A^{(1)}|S^{(1)}, \tau_{ps} = \tau, \phi_{ps} = \phi)P_{PS}(\tau_{ps} = \tau, \phi_{ps} = \phi|S^{(1)})d\tau d\phi$$

$$= p_{\tau',\phi'}(S^{(1)}) \int_{\tau} \pi^\star_{\tau,\phi}(A^{(1)}|S^{(1)})P_{PS}(\tau_{ps} = \tau, \phi_{ps} = \phi|S^{(1)})d\tau d\phi.$$

To continue, note that $P_{PS}(\tau_{ps} = \tau, \phi_{ps} = \phi|S^{(1)}) = P_{PS}(\tau_{ps} = \tau, \phi_{ps} = \phi|D)$ given that the posterior sampling does not depend on $S^{(1)}$. With (7), we further have

$$P_{PS}(\tau_{ps} = \tau, \phi_{ps} = \phi|D) = P(\tau, \phi|D).$$

Thus,

$$P_{PS}(S^{(1)}, A^{(1)}) = p_{\tau',\phi'}(S^{(1)}) \int_{\tau,\phi} \pi^\star_{\tau,\phi}(A^{(1)}|S^{(1)})P(\tau, \phi|D)d\tau d\phi$$

$$= p_{\tau',\phi'}(S^{(1)}) \int_{\tau,\phi} P(A^{(1)}|\tau, \phi, S^{(1)})P(\tau, \phi|D)d\tau d\phi$$

$$= p_{\tau',\phi'}(S^{(1)})P(A^{(1)}|S^{(1)}) = p_{\tau',\phi'}(S^{(1)})P_\theta(A^{(1)}|S^{(1)}) = P_\theta(S^{(1)}, A^{(1)}),$$

where the last line is due to and $T_\theta(A^{(1)}|S^{(1)}, D) = P(A^{(1)}|S^{(1)}, D)$ under Assumption E.2.

**Case $h$.** Assume that Case $h - 1$ holds that

$$P_{PS}(\Upsilon_{h-1}) = P_\theta(\Upsilon_{h-1}),$$

and we aims to prove $P_{PS}(\Upsilon_h) = P_\theta(\Upsilon_h)$, which is equivalent to prove

$$P_{PS}(S^{(h)}, A^{(h)}|\Upsilon_{h-1}) = P_\theta(S^{(h)}, A^{(h)}|\Upsilon_{h-1}),$$

because of the factorization

$$P_{PS}(S^{(h)}, A^{(h)}|\Upsilon_{h-1})P_{PS}(\Upsilon_{h-1}) = P_\theta(S^{(h)}, A^{(h)}|\Upsilon_{h-1})P_\theta(\Upsilon_{h-1}).$$

To this end, we have

$$P_{PS}(S^{(h)}, A^{(h)}|\Upsilon_{h-1}) = P_{PS}(S^{(h)}|\Upsilon_{h-1})P_{PS}(A^{(h)}|S^{(h)}, \Upsilon_{h-1}) = P_{\tau',\phi'}(S^{(h)}|\Upsilon_{h-1})P_{PS}(A^{(h)}|S^{(h)}, \Upsilon_{h-1})$$

$$= P_{\tau',\phi'}(S^{(h)}|\Upsilon_{h-1}) \int_{\tau,\phi} P_{PS}(A^{(h)}, \tau_{ps} = \tau, \phi_{ps} = \phi|S^{(h)}, \Upsilon_{h-1})d\tau d\phi$$

$$= P_{\tau',\phi'}(S^{(h)}|\Upsilon_{h-1}) \int_{\tau,\phi} P_{PS}(A^{(h)}, \tau_{ps} = \tau, \phi_{ps} = \phi|S^{(h)}, \Upsilon_{h-1})d\tau d\phi$$

$$= P_{\tau',\phi'}(S^{(h)}|\Upsilon_{h-1}) \int_{\tau,\phi} P_{PS}(A^{(h)}|\tau_{ps} = \tau, \phi_{ps} = \phi, S^{(h)}, \Upsilon_{h-1})P_{PS}(\tau_{ps} = \tau, \phi_{ps} = \phi|S^{(h)}, \Upsilon_{h-1})d\tau d\phi$$

$$= P_{\tau',\phi'}(S^{(h)}|\Upsilon_{h-1}) \int_{\tau,\phi} \pi^\star_{\tau,\phi}(A^{(1)}|S^{(1)})P_{PS}(\tau_{ps} = \tau, \phi_{ps} = \phi|S^{(h)}, \Upsilon_{h-1})d\tau d\phi.$$

To continue, we prove that $P_{PS}(\tau_{ps} = \tau, \phi_{ps} = \phi|S^{(h)}, \Upsilon_{h-1}) = P(\tau, \phi|S^{(h)}, \Upsilon_{h-1})$. Indeed,

$$P_{PS}(\tau, \phi|S^{(h)}, \Upsilon_{h-1}) = P_{PS}(\tau, \phi, S^{(h)}, \Upsilon_{h-1})/P_{PS}(S^{(h)}, \Upsilon_{h-1})$$

$$\propto P_{PS}(\tau, \phi, S^{(h)}, \Upsilon_{h-1})$$

$$= P_{PS}(\Upsilon_{h-1}|\tau, \phi)P_{PS}(S^{(h)}|\Upsilon_{h-1})P_{PS}(\tau, \phi|D)$$

$$\propto P(\tau, \phi|D)p_s(S^{(1:h)}) \prod_{h' \le h} \pi^\star_{\tau,\phi}(A^{(h')}|S^{(h')})$$

$$= P(\tau, \phi, S^{(h)}, \Upsilon_{h-1}) \propto P(\tau, \phi|S^{(h)}, \Upsilon_{h-1}).$$

Given that $P_{PS}(\tau, \phi | S^{(h)}, \Upsilon_{h-1})$ and $P(\tau, \phi | S^{(h)}, \Upsilon_{h-1})$ are distributions, $P_{PS}(\tau, \phi | S^{(h), \Upsilon_{h-1}}, \Upsilon_{h-1}) \propto P(\tau, \phi | S^{(h)}, \Upsilon_{h-1})$ implies that $P_{PS}(\tau, \phi | S^{(h)}, \Upsilon_{h-1}) = P(\tau, \phi | S^{(h)}, \Upsilon_{h-1})$. Thus,

$$
\begin{aligned}
P_{PS}(S^{(h)}, A^{(h)} | \Upsilon_{h-1}) &= P_{\tau', \phi'}(S^{(h)} | \Upsilon_{h-1}) \int_{\tau, \phi} \pi_{\tau, \phi}^{\star}(A^{(h)} | S^{(h)}) P(\tau, \phi | S^{(h)}, \Upsilon_{h-1}) d\tau d\phi \\
&= P_{\tau', \phi'}(S^{(h)} | \Upsilon_{h-1}) \int_{\tau, \phi} P(A^{(h)} | \tau, \phi, S^{(h)}, \Upsilon_{h-1}) P(\tau, \phi | S^{(h)}, \Upsilon_{h-1}) d\tau d\phi \\
&= P_{\tau', \phi'}(S^{(h)} | \Upsilon_{h-1}) P(A^{(h)} | S^{(h)}, \Upsilon_{h-1}) \\
&= P_{\tau', \phi'}(S^{(h)} | \Upsilon_{h-1}) P_\theta(A^{(h)} | S^{(h)}, \Upsilon_{h-1}) = P_\theta(S^{(h)}, A^{(h)} | \Upsilon_{h-1}),
\end{aligned}
$$

where the last line is due to Assumption E.2. Hence, the induction is complete and this concludes the proof. $\square$

### F.2 Proof of Theorem E.5

*Under Assumptions E.1, E.2 and E.4, in every iteration of ICAA (Algorithm 3), and for every variation environment $(\tau, \phi)$ with sufficient exploration $|D|$, the action label generation policy achieves performance no worse than the transformer policy $T_\theta$ from the previous iteration within the same environment $(\tau, \phi)$.*

*Proof of Theorem E.5.* We first prove a useful lemma.

**Lemma F.1.** *Under Assumptions E.4, when deploying a pretrained $T_\theta$ to a task $\tau^\star$ with an adversary $\phi^\star$, the performance of $T_\theta$ converges to that of the optimal policy, i.e.,*

$$
\mathbb{E}\left[\mathcal{R}_{\tau^\star}\left(T_\theta(\cdot | \cdot, D), \phi^\star\right)\right] \to \max_\pi \mathcal{R}_{\tau^\star}(\pi, \phi^\star) \quad as \quad |D| \to +\infty.
$$

*Proof of Lemma F.1.* As $|D| \to +\infty$, from Assumption E.4, we have the posterior $P(\tau, \phi | D)$ concentrates toward the truth $(\tau^\star, \phi^\star)$. As proved by Theorem E.3, a pretrained $T_\theta$ is performance Posterior Sampling during deployment. This leads to, for any neighbor $U(\epsilon)$ of $(\tau^\star, \phi^\star)$ with radius $\epsilon > 0$,

$$
\mathbb{E}\left[\mathcal{R}_{\tau^\star}\left(T_\theta(\cdot | \cdot, D), \phi^\star\right)\right] = \int P(\tau, \phi | D) \mathcal{R}_{\tau^\star}(\pi_{\tau, \phi}^\star, \phi^\star) \tag{8}
$$

$$
\geq P(U | D) \inf_{\tau', \phi' \in U(\epsilon)} \mathcal{R}_{\tau^\star}(\pi_{\tau', \phi'}^\star, \phi') \overset{|D| \to +\infty, \epsilon \to 0}{\longrightarrow} \mathcal{R}_{\tau^\star}(\pi_{\tau^\star, \phi^\star}^\star, \phi^\star) = \max_\tau \mathcal{R}_{\tau^\star}(\pi, \phi^\star). \tag{9}
$$

In addtion, by definition of $\mathcal{R}$. we have $\mathbb{E}\left[\mathcal{R}_{\tau^\star}\left(T_\theta(\cdot | \cdot, D), \phi^\star\right)\right] \leq \max_\tau \mathcal{R}_{\tau^\star}(\pi, \phi^\star)$ almost surely. This proves that $\mathbb{E}\left[\mathcal{R}_{\tau^\star}\left(T_\theta(\cdot | \cdot, D), \phi^\star\right)\right] \to \max_\tau \mathcal{R}_{\tau^\star}(\pi, \phi^\star)$. $\square$

For any $D$ of finite transitions, with Lemma F.1, we can outperform $T_\theta(\cdot | \cdot, D)$ by extending the exploration to have $D'$ where $|D'| > |D|$ such that the expected performance of $T_\theta(\cdot | \cdot, D')$ is arbitrarily close to the optimal one. Here, by definition of ICAA algorithm, $T_\theta(\cdot | \cdot, D')$ is the action label generation policy, thus concluding the proof. $\square$

## G Computation Requirements

We conduct each experiment on a single GPU: Nvidia RTX A5000 with 24GB RAM. In DarkRoom, all transformer models converge within 150 epochs within an hour. In Meta-World, we run each experiment up to 500 episodes with early termination mechanism. In MuJoCo, all models can converge within 200 epochs with less than two hours.

## H Pseudocode

**Algorithm 1** Deployment of ICRL Models

---

1: **Input:** Pretrained transformer Model $T_\theta$; Horizon of episodes $H$; Number of episodes $N$ for online testing; Offline dataset $D_{\text{off}} = \{(s_h, a_h, s_{h+1}, r_h)\}_h$, consisting of transitions collected by a behavioral policy.
2: // Offline Testing
3: **for** every time step $h \in \{1, \ldots, H\}$ **do**
4:     Observe state $s_h$
5:     Sample action with $T_\theta$:
$$a_h \sim T_\theta\left(\cdot | s_h, D_{\text{off}}\right)$$

6:     Collect reward $r_h$
7: **end for**
8: // Online Testing
9: Initialize an empty online data buffer $D_{\text{on}} = \{\}$
10: **for** every online trial $n \in \{1, \ldots, N\}$ **do**
11:     **for** every time step $h \in \{1, \ldots, H\}$ **do**
12:         Observe state $s_h$
13:         Sample action with $T_\theta$:
$$a_h \sim T_\theta\left(\cdot | s_h, D_{\text{on}}\right)$$

14:         Collect reward $r_h$
15:     **end for**
16:     Append the collected transitions $\{(s_h, a_h, s_{h+1}, r_h)\}_h$ into $D_{\text{on}}$
17: **end for**

---

# I Limitations

While ICAG involves generating expert policies across perturbed task variants, this step is performed efficiently in parallel and amortized over pretraining, making it feasible for a modest number of variants. ICAA uses self-generated labels derived from a robust pretrained model, which, despite not being optimal, have been shown empirically to improve performance with minimal data. These design choices strike a practical balance between robustness and scalability. While the proposed methods improve robustness under structured perturbations, their generalization is currently focused on tasks with in-distribution environment variations. Extending this framework to handle broader generalization—such as out-of-distribution task structures and disturbances—remains a valuable direction for future work.

**Algorithm 2** In-Context Adversarial Generalization
___
1: **Input:** Causal TM $T_\theta$; number of pretraining tasks $m$; number of variation environments per task $K$
2: //    Construction of Pretraining Dataset
3: Initialize an empty pretraining dataset $\mathcal{D}_{pre}$
4: **for** $i \in \{1, \ldots, m\}$ **do**
5:    Sample a task $\tau^i \in \mathcal{M}$;
6:    **for** $k \in \{1, \ldots, K\}$ **do**
7:       Sample a disturbance $\phi^{i,k} \in \Phi(\tau^i)$;
8:       Collect a context dataset $D^{i,k}$ from the variation environment $(\tau^i, \phi^{i,k})$;
9:       Sample a query state $s_{\text{query}}^{i,k}$;
10:      Train an optimal policy $\pi_{\tau^i, \phi^{i,k}}^\star$ following (5);
11:      Sample an action label $a_{i,k}^\star \sim \pi_{\tau^i, \phi^{i,k}}^\star(s_{\text{query}}^{i,k})$;
12:      Append $(D^{i,k}, s_{\text{query}}^{i,k}, a_{i,k}^\star)$ into $\mathcal{D}_{pre}$;
13:    **end for**
14: **end for**
15: //    Supervised Pretraining
16: Pretrain the TM $T_\theta$ by

$$\min_\theta \frac{1}{mK} \sum_{i=1}^{m} \sum_{k=1}^{K} -\log T_\theta(a_{i,k}^\star | s_{\text{query}}^{i,k}, D^{i,k}).$$

___

**Algorithm 3** In-Context Adversarial Adaptation
___
1: **Input:** number of pretraining tasks $m$; number of variation environments per task $K$; initial pretraining dataset $\mathcal{D}^0$
2: //    Initial Supervised Pretraining
3: Pretrain a causal transformer with $\mathcal{D}^0$ to have the pretrained TM $T_\theta^0$.
4: Set $\mathcal{D}_{pre} = \mathcal{D}^0$.
5: **for** $j \in \{0, \ldots, J\}$ **do**
6:    //    Collecting New Pretraining Data
7:    Initialize an empty dataset $\mathcal{D}^{j+1}$ for new data.
8:    **for** $i \in \{1, \ldots, m\}$ **do**
9:      Sample a task $\tau^i \in \mathcal{M}$;
10:     **for** $k \in \{1, \ldots, K\}$ **do**
11:       Sample a disturbance $\phi^{i,k} \in \Phi(\tau^i)$;
12:       Deploy $T_\theta^j$ for each variation environment $(\tau^i, \phi^{i,k})$ (following Algorithm 1) with $N+1$ trials to have trajectories $\xi_n = \{(s_h, a_h, s_{h+1}, r_h)\}_h, n \in \{0, \cdots, N\}$.
13:       Use the first $\underline{N}+1$ trajectories as context datasets $D_n = \xi_n, n \in \{0, \cdots, \underline{N}\}$.
14:       Sample query state-action label pairs from the remaining trajectories: $\{(s_{\text{query}}^n, a^n)\}_{n=0}^{N} \sim$ $\bigcup_{n=\underline{N}}^{N} \{(s_h, a_h) \in \xi_n\}_{h=0}^{H-1}$
15:       Append $\{D_n, s_{\text{query}}^n, a^n\}_{n=0}^{N}$ into $\mathcal{D}^{j+1}$;
16:     **end for**
17:    **end for**
18:    //    Supervised Fine-Tuning
19:    Update the pretraining dataset with the new data $\mathcal{D}_{pre} = \mathcal{D}_{pre} \cup \mathcal{D}^{j+1}$.
20:    Pretrain the latest TM $T_\theta^j$ with the updated $\mathcal{D}_{pre}$ to have $T_\theta^{j+1}$.
21: **end for**

