# OpenReview forum: "Efficiently Robust In-Context Reinforcement Learning with Adversarial Generalization and Adaptation"
_NeurIPS.cc/2025/Workshop/Reliable_ML — NeurIPS 2025 - Reliable ML Workshop_

### Official Review · Reviewer_592j · 2025-09-17
**Review: Missing Analysis of Model Scale and Architectural Generalization**

**Rating:** 7
**Confidence:** 4

**Review:**

Paper summary: In In-Context Reinforcement Learning (ICRL), when pretrained transformer models are deployed in new environments that have perturbations, their performance is significantly degraded. This paper addresses this challenge by proposing the augmentation of pretraining data with adversarial variations. Their goal is to improve generalization without sacrificing performance, and as such, they work to find the optimal policy for each specific variation environment and use it for pretraining. This allows the TM to generalize across multiple disturbances and adversaries. They also address the computing overhead problem of finding actions for the variation environments without the need to train from scratch by using online rollouts of the ICRL to collect optimal action labels for each variation.

Strengths: This paper departs from traditional, computationally expensive, worst-case optimization methods. The transformation of the worst-case scenario problem into a generalization problem using ICAG is insightful. The work also presents comprehensive experimental results across a diverse set of challenging environments (sparse-reward navigation, robotic manipulation, continuous control). The experiments rigorously compare ICAG and ICAA against multiple strong baselines, consistently demonstrating their superior performance under various disturbances and out-of-distribution conditions. The ablation studies on the number of adversaries and ICAA iterations further strengthen the empirical claims. The connection of ICAG to implicit Posterior Sampling offers a deep insight into its sample efficiency and optimal adaptation capability. The paper is well-written with clear explanations of complex concepts and algorithms. The core problem addressed, which is enhancing the robustness of ICRL models to disturbances, is directly aligned with the workshop's theme. These disturbances can be seen as corrupted or unreliable interaction data.

Weaknesses/ Limitations:
Missing Analysis of Model Scale and Architecture: The paper focuses on a GPT-2 transformer but does not provide a comprehensive analysis of how its proposed frameworks would perform with larger models (e.g., GPT-2 XL, or modern architectures like Llama or DeepSeek). This is a significant limitation, as scaling is a critical factor in the performance and efficiency of in-context learning methods.

Suggestions:
1. Conduct Scaling Experiments: The authors should investigate the effects of model size on their proposed frameworks, In-context Adversarial Generalization (ICAG) and In-context Adversarial Adaptation (ICAA). This could involve running experiments on a range of model sizes to see how the robustness and generalization benefits scale.

2. Explore Different Architectures: To strengthen the paper's findings and demonstrate broader applicability, the authors could test their frameworks on different transformer architectures, such as Llama or DeepSeek. This would provide valuable insights into how the specific architectural choices (e.g., Mixture-of-Experts in DeepSeek) interact with their proposed methods.

3. Analyze Data Scaling: An ablation study on the size and diversity of the training data would be a valuable addition. Investigating how the performance of the proposed methods changes as the amount of adversarial data is varied could provide a deeper understanding of their underlying mechanisms.

---

### Official Review · Reviewer_JNir · 2025-09-20

**Rating:** 7
**Confidence:** 2

**Review:**

* Summary

The paper introduces two methods for improving robustness in in-context RL: (1) In-Context Adversarial Generalization (ICAG), which augments pretraining with adversarially perturbed task variants, and (2) In-Context Adversarial Adaptation (ICAA), which adapts models through online rollouts. Experiments on Dark Room, Meta-World, and MuJoCo show improved robustness compared to DPT-based baselines.

* Strengths

Addresses an important gap: robustness of transformer-based ICRL under perturbations. ICAG reframes robustness as generalization, avoiding pessimism from max–min optimization. ICAA adds a practical self-improvement loop with limited extra training cost. Results are consistent across several benchmarks.

* Weaknesses / Limitations

Baseline coverage is limited: lacks comparison with more advanced robust/adversarial RL or domain randomization methods. In addition, creating many adversarial variants (ICAG) may still be expensive in large-scale settings.

* Suggestions for Authors

Expand empirical comparisons to include stronger baselines from robust RL and domain randomization. Provide clearer discussion of computational trade-offs at scale.

* Ethics

No major ethical issues. A short discussion of adversarial robustness in safety-critical applications would be valuable.